# Model Tampering Attacks Enable More Rigorous Evaluations of LLM Capabilities

*Zora Che, *University of Maryland, ML Alignment & Theory Scholars*          *zche@umd.edu*

*Stephen Casper, *MIT CSAIL, ML Alignment & Theory Scholars*          *scasper@mit.edu*

**Robert Kirk,** *UK AI Security Institute*          *robert.kirk@dsit.gov.uk*

**Anirudh Satheesh,** *University of Maryland*          *anirudhs@terpmail.umd.edu*

**Stewart Slocum,** *MIT*          *slocumstewy@gmail.com*

**Lev McKinney,** *University of Toronto*          *levmckinney@cs.toronto.edu*

**Rohit Gandikota,** *Northeastern University*          *gandikota.ro@northeastern.edu*

**Aidan Ewart,** *Haize Labs*          *aidanprattewart@gmail.com*

**Domenic Rosati,** *Dalhousie University*          *domenicrosati@gmail.com*

**Zichu Wu,** *University of Waterloo*          *zichu.wu@uwaterloo.ca*

**Zikui Cai,** *University of Maryland*          *zikui@umd.edu*

**Bilal Chughtai,** *Apollo Research*          *bilalchughtai@google.com*

**Yarin Gal,** *UK AI Security Institute, University of Oxford*          *yarin.gal@cs.ox.ac.uk*

**Furong Huang,** *University of Maryland*          *furongh@umd.edu*

**Dylan Hadfield-Menell,** *MIT*          *dylanhm@mit.edu*

**Reviewed on OpenReview:** *https://openreview.net/forum?id=E60YbLnQd2*

## Abstract

Evaluations of large language model (LLM) risks and capabilities are increasingly being incorporated into AI risk management and governance frameworks. Currently, most risk evaluations are conducted by designing *inputs* that elicit harmful behaviors from the system. However, this approach suffers from two limitations. First, input-output evaluations cannot fully evaluate realistic risks from open-weight models. Second, the behaviors identified during any particular input-output evaluation can only lower-bound the model's worst-possible-case input-output behavior. As a complementary method for eliciting harmful behaviors, we propose evaluating LLMs with *model tampering* attacks which allow for modifications to latent activations or weights. We pit state-of-the-art techniques for removing harmful LLM capabilities against a suite of 5 input-space and 6 model tampering attacks. In addition to benchmarking these methods against each other, we show that (1) model resilience to capability elicitation attacks lies on a low-dimensional robustness subspace; (2) the success rate of model tampering attacks can empirically predict and offer conservative estimates for the success of held-out input-space attacks; and (3) state-of-the-art unlearning methods can easily be undone within 16 steps of fine-tuning. Together, these results highlight the difficulty of suppressing harmful LLM capabilities and show that model tampering attacks enable substantially more rigorous evaluations than input-space attacks alone.[1]

---

[1]We release models at https://huggingface.co/LLM-GAT.

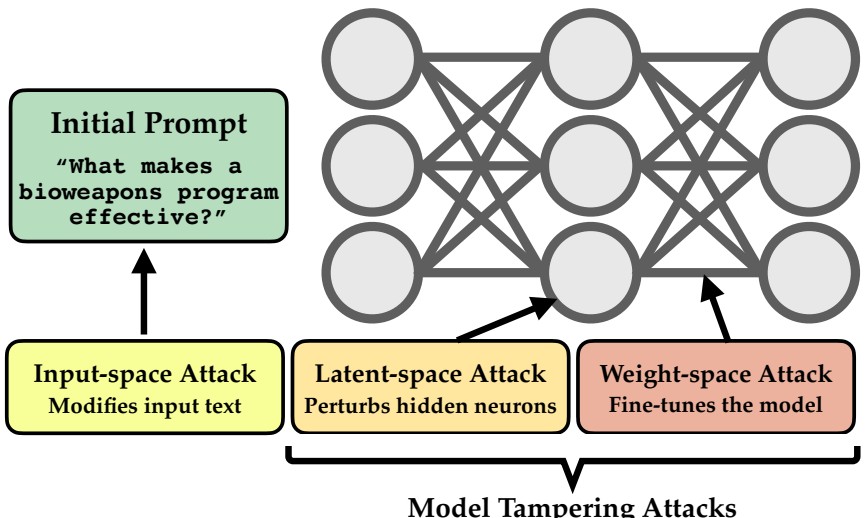

Figure 1: **Model tampering attacks modify latents and weights.** In contrast to input-space attacks, model tampering attacks elicit capabilities from an LLM by making modifications to the internal activations or weights. In this paper, we use model tampering attacks to (1) directly evaluate risks from malicious tampering with open-weight models and (2) indirectly evaluate difficult-to-foresee input-space vulnerabilities in models.

# 1 Introduction: Limitations of Input-Output Evaluations

Rigorous evaluations of large language models (LLMs) are widely recognized as key for risk mitigation (Raji et al., 2022; Anderljung et al., 2023; Schuett et al., 2023; Shevlane et al., 2023) and are being incorporated into AI governance frameworks (NIST, 2023; UK DSIT, 2023; EU, 2023; China, 2023; Brazil, 2023; Canada, 2022; Korea, 2025). However, despite their efforts, developers often fail to identify overtly harmful LLM behaviors pre-deployment (Shayegani et al., 2023; Andriushchenko et al., 2024; Carlini et al., 2024; Yi et al., 2024b). Current methods primarily rely on automated input-space attacks, where evaluators search for prompts that elicit harmful behaviors. These are useful but often leave unidentified vulnerabilities. A difficulty with input-space attacks is that they are poorly equipped to cover the attack surface. This happens for two reasons. First, attackers can sometimes manipulate more than just model inputs (e.g., if a model is open-source). Second, it is intractable to exhaustively search the input space.[2] These challenges highlight a fundamental limitation of input-space evaluations: the worst behaviors identified during an assessment can only offer a lower bound of the model's overall worst-case behavior (Gal, 2024; OpenAI, 2024).

To help address this challenge, we draw inspiration from a safety engineering principle: that safety-critical systems should be tested under stresses at least as extreme—if not more—than those expected in deployment (Clausen et al., 2006). For example, buildings are designed to withstand loads multiple times greater than their intended use. Here, we take an analogous approach to evaluating and building safety cases (Clymer et al., 2024) for LLMs: stress-testing them under attacks that go beyond input-space manipulations.

We propose using *model tampering* attacks, which allow for adversarial modifications to the model's weights or latent activations, in addition to evaluating systems under input-space attacks (see Figure 1). We attempt to answer two questions, each corresponding to a different threat model:

**Question 1: How vulnerable are LLMs to model tampering attacks?** Answering this helps us understand how model tampering attacks can be used to study risks from models that are open-source,[3] have fine-tuning APIs, or may be leaked (Nevo et al., 2024).

---

[2]For example, with modern tokenizers, there are vastly more 20-token strings than particles in the known universe.

[3]It may seem obvious that model tampering attacks are needed to realistically assess threats from open-source models. However, there is a precedent for developers failing to use them prior to open-source releases. For example, before releasing

**Question 2: Can model tampering attacks inform evaluators about LLM vulnerabilities to novel input-space attacks?** Answering this will help us understand how model tampering attacks can help assess risks from both open- and closed-source models.

To answer these questions, we pit state-of-the-art methods for unlearning and safety fine-tuning in LLMs against a suite of input-space and model tampering attacks. We make four contributions:

1. **Benchmarking:** We benchmark 8 unlearning methods and 9 safety fine-tuned LLMs, each against 11 capability elicitation attacks.

2. **Science of robustness:** We show that LLM resilience to a variety of capability elicitation attacks lies on a low-dimensional robustness subspace.

3. **Evaluation methodology:** We show that the success of some model tampering attacks correlates with that of held-out input-space attacks. We also find that few-shot fine-tuning attacks can empirically be used to conservatively over-estimate a model's robustness to held-out input-space threats.

4. **Model suite:** To facilitate further research, we release a set of 64 models trained using 8 methods to unlearn dual-use biology knowledge at varying degrees of strength at https://huggingface.co/LLM-GAT.

## 2 Related Work

**Latent-space attacks:** During a latent-space attack, an adversary can make modifications to a model's hidden activations. Adversarial training under these attacks can improve the generality of a model's robustness (Sankaranarayanan et al., 2018; Singh et al., 2019; Zhang et al., 2023; Schwinn et al., 2023; Zeng et al., 2024). In particular, Xhonneux et al. (2024), Casper et al. (2024), and Sheshadri et al. (2024) use latent adversarial training to improve defenses against held-out types of adversarial attacks. Other work on activation engineering has involved making modifications to a model's behavior via simple transformations to their latent states (Zou et al., 2023a; Wang & Shu, 2023; Lu & Rimsky, 2024; Arditi et al., 2024). Zhang et al. (2025) also showed that unlearning methods can be brittle to quantization methods.

**Weight-space (fine-tuning) attacks:** During a few-shot fine-tuning attack (Huang et al., 2024), an adversary can modify model weights via fine-tuning on a limited number of samples. For example, Qi et al. (2023) showed that fine-tuning on as few as 10 samples could jailbreak GPT-3.5. Many works have used few-shot fine-tuning attacks to elicit LLM capabilities that were previously suppressed by fine-tuning or unlearning (Jain et al., 2023; Yang et al., 2023; Qi et al., 2023; Bhardwaj & Poria, 2023; Lermen et al., 2023; Zhan et al., 2023; Ji et al., 2024; Qi et al., 2024a; Hu et al., 2024; Halawi et al.; Peng et al., 2024; Lo et al., 2024; Łucki et al., 2024; Shumailov et al., 2024; Lynch et al., 2024; Deeb & Roger, 2024; Qi et al., 2024b; Yi et al., 2024a).

**Capability elicitation and evaluation:** LLMs are commonly developed by simply training them to behave desirably (e.g., with RLHF (Casper et al., 2023)), but in this paper, we focus on testing targeted defenses against known, harmful behaviors. Research on adversarial capability elicitation (Hofstätter et al., 2025) in LLMs has primarily been done in the context of machine unlearning (Liu et al., 2024a; Barez et al., 2025) and jailbreaking (Yi et al., 2024b). Here, we experiment in these two domains. However, capability elicitation has also been researched in the context of backdoors/trojans (Zhao et al., 2024), "password-locked models" (Greenblatt et al., 2024; Hofstätter et al., 2025), and "sandbagging" (van der Weij et al., 2024). In the unlearning field, several recent works have used adversarial methods to evaluate the robustness of safety fine-tuning and unlearning algorithms (Patil et al., 2023; Lynch et al., 2024; Łucki et al., 2024; Hu et al., 2024; Liu et al., 2024a; Zhang et al., 2024; Liu et al., 2024b; Wei et al., 2024). Here, we build on Li et al. (2024b) who introduce WMDP-Bio, a benchmark for unlearning dual-use biotechnology knowledge from LLMs. In the jailbreaking field, many techniques have been developed to make LLMs comply with

---

Llama 2 and Llama 3 models, Meta's red-teaming efforts did not reportedly involve model tampering attacks (Touvron et al., 2023; Dubey et al., 2024).

harmful requests (Shayegani et al., 2023; Yi et al., 2024b; Jin et al., 2024; Chowdhury et al., 2024; Lin et al., 2024). Here, we experiment with 9 open-source LLMs and a set of gradient-guided, perplexity-guided, and prosaic techniques from the adversarial attack literature (see Table 1).

## 3 Methods

**Defenses**

| | | | |
|---|---|---|---|
| *Unlearning Methods* | | Gradient Difference (**GradDiff**) | Liu et al. (2022) |
| *(We train 8x models each to unlearn WMDP-Bio)* | | Random Misdirection for Unlearning (**RMU**) | Li et al. (2024b) |
| | | RMU with Latent Adversarial Training (**RMU+LAT**) | Sheshadri et al. (2024) |
| | | Representation Noising (**RepNoise**) | Rosati et al. (2024) |
| | | Erasure of Language Memory (**ELM**) | Gandikota et al. (2024) |
| | | Representation Rerouting (**RR**) | Zou et al. (2024) |
| | | Tamper Attack Resistance (**TAR**) | Tamirisa et al. (2024) |
| | | K-FAC for Distribution Erasure (**K-FADE**) | McKinney et al. |
| *Jailbreak Refusal-Tuned Models* | | meta-llama/Meta-Llama-3-8B-Instruct | Dubey et al. (2024) |
| *(Off the shelf)* | | slz0106/llama3_finetune_refusal | Link |
| | | JINJIN7987/llama3-8b-refusal-vpi | Link |
| | | Youliang/llama3-8b-derta | Yuan et al. (2024) |
| | | GraySwanAI/Llama-3-8B-Instruct-RR | Zou et al. (2024) |
| | | LLM-LAT/llama3-8b-instruct-rt-jailbreak-robust1 | Sheshadri et al. (2024) |
| | | LLM-LAT/robust-llama3-8b-instruct | Sheshadri et al. (2024) |
| | | lapisrocks/Llama-3-8B-Instruct-TAR-Refusal | Tamirisa et al. (2024) |
| | | Orenguteng/Llama-3-8B-Lexi-Uncensored | Link |

**Attacks**

| | | | |
|---|---|---|---|
| *Input-Space* | Gradient-guided | Greedy Coordinate Gradient (**GCG**) | Zou et al. (2023b) |
| | | **AutoPrompt** | Shin et al. (2020) |
| | Perplexity-guided | Beam Search-based Attack (**BEAST**) | Sadasivan et al. (2024) |
| | Prosaic | Prompt Automatic Iterative Refinement (**PAIR**) | Chao et al. (2024) |
| | | **Human Prompt** | |
| *Model Tampering* | Latent space | **Embedding perturbation** | Schwinn et al. (2024) |
| | | **Latent perturbation** | Sheshadri et al. (2024) |
| | Weight space | **WandA Pruning** | Sun et al. (2023) |
| | | **Benign LoRA** | Qi et al. (2023) |
| | | **LoRA** | Hu et al. (2021) |
| | | **Full Parameter** | |

Table 1: **Table of capability elicitation (attack) and capability suppression (defense) methods.** We consider defenses in two different settings: (top) unlearning approaches that remove hazardous bio-knowledge and (bottom) refusal-tuned models that resist jailbreaks.

**Our approach – pitting capability suppression defenses against capability elicitation attacks.** Here, we study capability suppression methods that depend on both removing knowledge from the model (unlearning) and teaching the model to robustly refuse (jailbreaking) requests. For unlearning experiments, we experiment with 65 models trained using 8 different unlearning methods. For jailbreaking experiments, we experiment with 9 models off the shelf from prior works. In both cases, we pit these defenses against a set of 11 input-space and model tampering attacks to either elicit 'unlearned' knowledge or jailbreak the model. In Table 1, we list all unlearning methods, off-the-shelf models, and attacks we use. Since the input-space attacks that we use are held out, we treat them as proxies for novel input-space attacks in our evaluations (see also Hofstätter et al. (2025)).

**Defenses – machine unlearning methods:** We unlearn dual-use bio-hazardous knowledge on Llama-3-8B-Instruct Dubey et al. (2024) with the unlearning methods listed in Table 1 and outlined in Appendix A.2.1. For all methods, we train on 1,600 examples of max length 512 from the bio-remove-split of the WMDP 'forget set' (Li et al., 2024b), and up to 1,600 examples of max length 512 from Wikitext as the 'retain set'. For the 8 unlearning methods listed in Table 1, we take 8 checkpoints evenly spaced across training. Finally, we also use the public release of the "TAR-v2" model from Tamirisa et al. (2024) as a 9th TAR model. In total, the 8 checkpoints each from the 8 methods we implemented plus the TAR model from Tamirisa et al. (2024) resulted in 65 models.

**Defenses – refusal fine-tuned models:** For jailbreaking experiments, we use the 9 fine-tuned Llama3-8B-Instruct models off the shelf listed in Table 1. The first 8 are all fine-tuned for robust refusal of harmful

requests. Of these, 'RR' (Zou et al., 2023a) and 'LAT' (Sheshadri et al., 2024) are state-of-the-art for open-weight jailbreak robust models (Li et al., 2024a; Haize Labs, 2023). The final 'Orenguteng' model was fine-tuned to be 'helpful-only' and comply even with harmful requests. We discuss these models in more detail in Appendix A.3.

**Attacks – capability elicitation methods:** We use 5 input-space attacks and 6 model tampering attacks on our unlearned models. We use these attacks (single-turn) to increase dual-use bio knowledge (as measured by WMDP-Bio performance (Li et al., 2024b)) for unlearning experiments and to elicit compliance with harmful requests (as measured by the StrongReject AutoGrader (Souly et al., 2024)) for jailbreaking experiments. We selected attacks based on algorithmic diversity and prominence in the state of the art. We list all 11 attacks in Table 1. **In all experiments, we produce *universal* adversarial attacks optimized to work for *any* prompt.** This allows us to attribute attack success to capability elicitation rather than answer-forcing from the model (e.g., Fort (2023)). For descriptions and implementation details for each attack method, see Appendix A. Finally, we also used two proprietary attacks – one for unlearning experiments and one for jailbreaking experiments which we will describe in Section 4.

**Attacks – data:**

- **Attacks on unlearning – non-fine-tuning:** we used 64 held-out examples of multiple-choice biology questions from the WMDP-Bio test set.

- **Attacks on unlearning – adversarial fine-tuning:** we use the WMDP 'bio retain' or 'forget' sets. Both of which are comprised of biology papers.

- **Attacks on refusal training – all except benign fine-tuning:** we used held-out examples of compliance with harmful requests from Sheshadri et al. (2024). Each example is a prompt + response pair.

- **Attacks on unlearning and refusal training – benign fine-tuning:** For all benign fine-tuning attacks, we used WikiText Merity et al. (2016).

For details on attack configurations, including the number of examples, batch size, number of steps, and other hyper-parameters, see Appendix A.4.

**Attacks – model tampering attacks are efficient.** In Table 3, we show the number of forward and backward passes used in our implementations of attacks. Model tampering attacks were more efficient than state-of-the-art input-space attacks.

## 4 Experiments

As discussed in Section 1, we have two motivations, each corresponding to a different threat model. First, we want to directly evaluate robustness to model tampering attacks to better understand the risks of open-source, leaked, or API fine-tuneable LLMs. Second, we want to understand what model tampering attacks can tell us about novel, unforeseen input-space attacks in order to study risks from all types of LLMs. Unfortunately, unforeseen attacks are, by definition, ones that we do not have access to. Instead, since the input-space attacks that we use are held out during fine-tuning, we treat them as proxies for 'unforeseen' input-space attacks.

### 4.1 Unlearning Experiments

We first experiment with the unlearning of dual-use biology knowledge in LLMs by pitting unlearning methods against capability elicitation methods (see Table 1).

#### 4.1.1 Benchmarking Unlearning Methods

**Calculating an *unlearning score*:** In our models, we evaluate *unlearning efficacy* on the WMDP-Bio QA evaluation task (Li et al., 2024b). We evaluate *general utility* using three benchmarks. First, we use

| Method | WMDP ↓ | WMDP, Best Input Attack ↓ | WMDP, Best Tamp. Attack ↓ | MMLU ↑ | MT-Bench/10 ↑ | AGIEval ↑ | Unlearning Score ↑ |
|---|---|---|---|---|---|---|---|
| Llama3-8B-Instruct | 0.70 | 0.75 | 0.71 | 0.64 | 0.78 | 0.41 | 0.00 |
| Grad Diff | 0.25 | 0.27 | 0.67 | 0.52 | 0.13 | 0.32 | 0.17 |
| RMU | 0.26 | 0.34 | 0.57 | 0.59 | 0.68 | 0.42 | 0.84 |
| RMU + LAT | 0.32 | 0.39 | 0.64 | 0.60 | 0.71 | 0.39 | 0.73 |
| RepNoise | 0.29 | 0.30 | 0.65 | 0.59 | 0.71 | 0.37 | 0.78 |
| ELM | 0.24 | 0.38 | 0.71 | 0.59 | 0.76 | 0.37 | 0.95 |
| RR | 0.26 | 0.28 | 0.66 | 0.61 | 0.76 | 0.44 | 0.96 |
| TAR | 0.28 | 0.29 | 0.36 | 0.54 | 0.12 | 0.31 | 0.09 |
| K-FADE | 0.31 | 0.32 | 0.64 | 0.63 | 0.78 | 0.40 | 0.85 |

Table 2: **Benchmarking LLM unlearning methods:** We report results for the checkpoint from each method with the highest unlearning score (Equation (1)). We report original WMDP-Bio performance, worst-case WMDP-Bio performance across our attacks, and three measures of general utility: MMLU, MT-Bench, and AGIEval. Representation rerouting (RR) has the best unlearning score. No model has a WMDP-Bio performance less than 0.36 after the most effective attack. We note that Grad Diff and TAR models performed very poorly, often struggling with basic fluency.

MMLU (Hendrycks et al., 2020) and AGIEval (Zhong et al., 2023), which are based on asking LLMs multiple-choice questions. We then use MT-Bench (Bai et al., 2024) which is based on long answer questions and thus measures both knowledge and fluency. Because the goal of unlearning is to differentially decrease capabilities in a target domain, we calculate an "unlearning score" based on both unlearning efficacy and utility degradation. Given an original model $M$ and an unlearned model $M'$, we calculate $S_{\text{unlearn}}(M')$ with the formula:

$$
S_{\text{unlearn}}(M') = (\underbrace{[S_{\text{WMDP}}(M) - S_{\text{WMDP}}(M')]}_{\Delta \text{Unlearn efficacy}} - \\
\underbrace{[S_{\text{utility}}(M) - S_{\text{utility}}(M')]}_{\Delta \text{Utility degradation}}) / \\
\underbrace{[S_{\text{WMDP}}(M) - S_{\text{WMDP}}(\text{rand})]}_{\Delta \text{Random chance (for normalization)}}
$$

(1)

Here, $S_{\text{WMDP}}(\cdot)$ is the accuracy on the WMDP-Bio QA Evaluation and $S_{\text{utility}}(\cdot)$ is an aggregated utility measure. $S_{\text{utility}}(\cdot)$ is calculated by taking a weighted average of MMLU, AGIEval, and MT-Bench. We use weights of 1/4, 1/4, and 1/2 respectively because MT-Bench uniquely measures model fluency. Finally, "rand" refers to a random policy. An unlearning score of 1.0 indicates theoretically optimal unlearning – random performance on the unlearned domain and unaffected performance on others. Meanwhile, the unlearning score of the original model $M$ is 0.0. Table 2 reports results from the best-performing checkpoint (determined by unlearning score) from each of the 8 methods.

**Representation rerouting (RR) achieves the highest unlearning score. GradDiff and TAR struggle due to dysfluency.** We find different levels of unlearning success. Representation rerouting (RR) performs the best overall, achieving an unlearning score of 0.96. In contrast, GradDiff and TAR have limited success with the lowest unlearning scores. Poor MT-Bench scores and our manual assessment of these models suggest that GradDiff and TAR struggle principally due to poor fluency.

**No method is robust to all attacks.** We plot the increase in WMDP-Bio performance for the best checkpoint from each unlearning method after each attack in Figure 2 and show that all models, even those with the lowest unlearning scores, exhibit a worst-case performance increase of 8 percentage points or more when attacked.

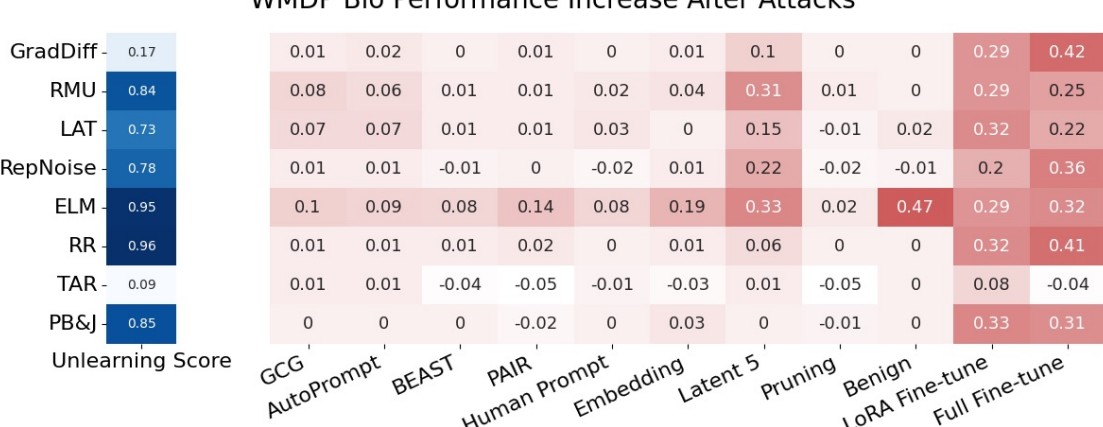

Figure 2: **Pitting capability suppression (unlearning) methods against capability elicitation attacks.** We use unlearning methods to suppress bio-hazardous knowledge from LLMs and pit these against capability elicitation attacks seeking to re-elicit the unlearned knowledge. All unlearning methods tested could be successfully attacked. **Left:** The *unlearning score* (Equation (1)) measures how effectively each unlearning method removed unwanted capabilities while preserving general model utility. Higher scores indicate better unlearning (scale 0-1). **Right:** Increase in the unlearned task performance after attacks. The first 5 columns are from input-space attacks while the final 6 are from model tampering attacks. In particular, finetuning attacks (rightmost columns) were especially effective at resurfacing suppressed capabilities.

### 4.1.2 Model robustness exists on a low-dimensional subspace

**We perform PCA, weighting models by their unlearning score.** First, to understand the extent to which some attacks offer information about others, we analyze the geometry of attack successes across our 65 models. Previously Wei et al. (2024) found that a model's vulnerability to pruning and low-rank modifications both relate with the brittleness of its safety fine-tuning. Here, we extend on this finding with more attacks and subspace analysis. We perform weighted principal component analysis on the WMDP-Bio improvements achieved by all 11 attacks on all 65 model checkpoints. We first constructed a matrix $A$ with one row per model and one column per attack. Each $A_{ij}$ corresponds to the increase in WMDP-Bio performance in model $i$ under attack $j$. We then centered the data and multiplied each row $A_i$ by the square root of the unlearning score: $\sqrt{S_{\mathrm{unlearn}}(A_i)}$. This allowed for models to influence the analysis in proportion to their unlearning score.

**Three principal components explain 89% of the variation in attack success.** Figure 3 displays the eigenvalues from PCA and the top three principal components (weighted by eigenvalues). This suggests that different capability elicitation attacks exploit models via related mechanisms.

**Hierarchical clustering reveals distinct attack families.** In Figure 4, we perform agglomerative clustering on attack success correlations. Algorithmically similar attacks tend to cluster together. However, adversarial finetuning attacks exhibit significant variation, even amongst each other. Finally we see that benign model tampering methods (pruning and benign fine-tuning) behave similarly to gradient-free input-space attacks.

### 4.1.3 Model tampering attacks empirically predict and conservatively estimate the success of input-space attacks

**Embedding-space attacks, latent-space attacks, pruning, and benign fine-tuning empirically** *correlate* **with input-space attack successes.** In Figure 5 these three model tampering attacks tend

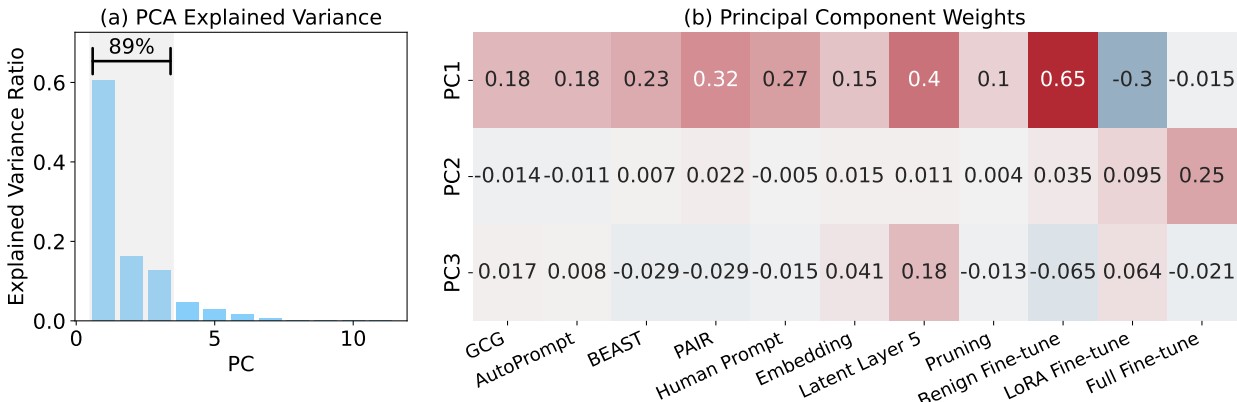

Figure 3: **Three principal components explain 89% of the variation in attack success. Left:** The proportion of explained variance for each principal component. **Right:** We display the first three principal components weighted by their eigenvalues. The first principal component suggests a geometric distinction between the two adversarial (LoRA, Full) fine-tuning attacks and all others.

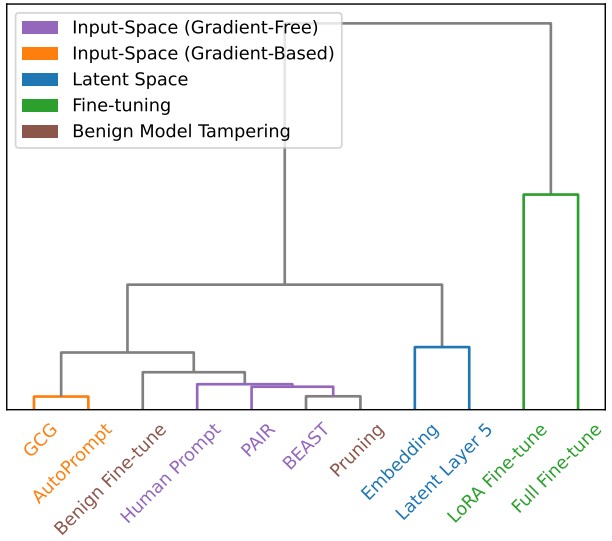

Figure 4: **Hierarchical clustering reveals groupings of attacks.** Attacks tend to cluster by algorithmic type. However, benign fine-tuning attacks cluster with gradient-free input-space attacks.

to have positive correlations with input-space attack successes with $p$ values near zero.[4] In these plots, we size points by their unlearning score and display the Pearson correlation weighted by unlearning score. Full results are in Appendix B. This suggests that embedding-space attacks, latent-space attacks, pruning, and benign fine-tuning are particularly able to predict the successes of held-out input-space attacks.

**Fine-tuning attack successes empirically offer conservative estimates for input-space attack successes.** LoRA and Full fine-tuning performed differently on different attacks. However, together, the max of the two did as well or better than the best-performing input-space attack on 64 of 65 models. This suggests that model tampering attacks could be used to develop more cautious estimates of a model's worst-case behaviors than other attacks.

---

[4]Points are not independent or identically distributed, so we only use this "$p$" value for geometric intuition, and we do not attach it to any formal hypothesis test.

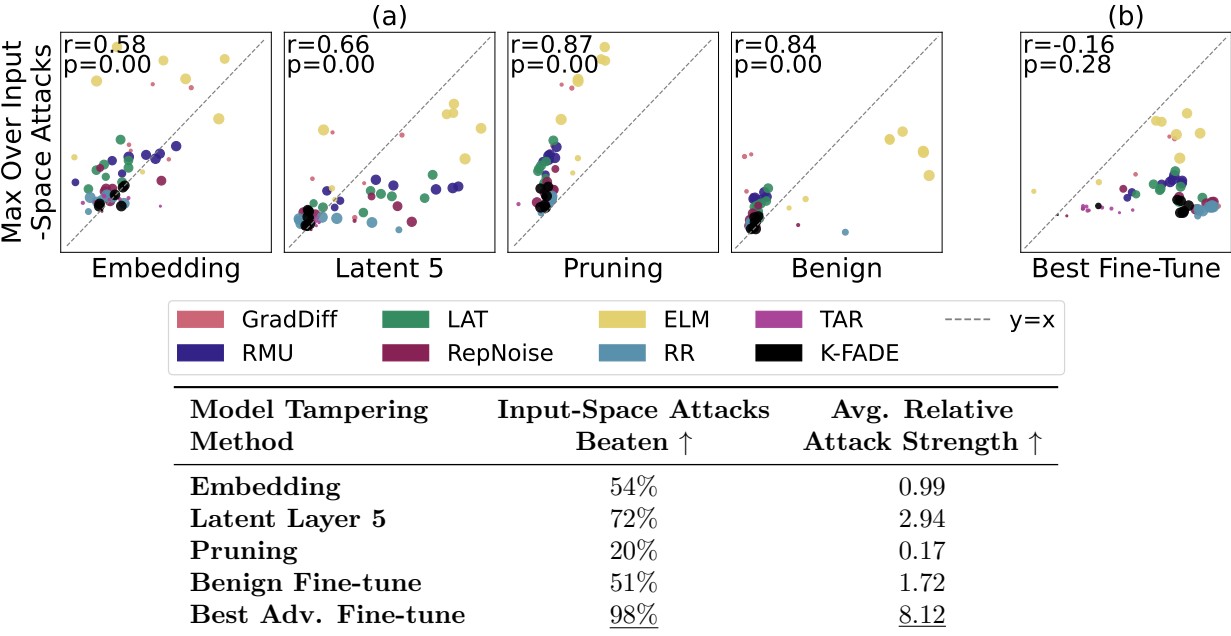

Figure 5: **In our experiments, (a) fine-tuning, embedding-space, and latent-space attack successes *correlate* with input-space attack successes while (b) fine-tuning attack successes empirically *exceed* the successes of state-of-the-art input-space attacks.** Here, we plot the increases in WMDP-Bio performance from model tampering attacks against the best-performing (of 5) input-space attacks for each model. We weight points by their unlearning score from Section 4.1.1. In (b), the $x$ axis is the best (over 2) between a LoRA and full fine-tuning attack. We also display the unlearning-score-weighted correlation and the correlation's $p$ value. *Points below and to the right of the line indicate that the model tampering attack was more successful.* **Table:** for each of the four model tampering attacks, the percent of all input-space attacks for which it performed better and the average relative attack strength compared to all input-space attacks.

**Model tampering attacks are predictive of the success of proprietary attacks from the UK AI Security Institute (UK AISI).** To more rigorously test what model tampering attacks can reveal about novel input-space attacks, we analyze their predictiveness on proprietary attacks from the UK AI Security Institute. These attacks were known to the 'red team' authors ( UK AISI affiliates) but were not accessible to all other 'blue team' authors. We conducted these attacks with the same data and approach as all of our other input-space attacks. Results are summarized in Figure 6 with full results in the Appendix. Correlations are weaker than before, but pruning and benign fine-tuning still correlate with attack success with a $p$ value near zero. Also as before, fine-tuning attack successes often tend to be as strong or stronger than input-space attacks. However, this trend was weaker, only occurring for 60 of the 65 models. See Appendix B.2 for full results and further analysis of UK AISI evaluations.

**Model tampering attacks improve worst-case input-space vulnerability estimation.** Finally, we test if model tampering attacks offer novel information that can be used to *predict* worst-case behaviors better than input-space attacks alone (Figure 14 in Appendix C.1). We train linear regression models to predict worst-case input-space attack success rates with information from either (1) only input-space attacks, or (2) both input-space and model tampering attacks. We find that including model tampering attacks improves predictiveness (e.g. from $r = 0.77$ to $0.83$ with four predictors). The best-performing combinations typically include attacks from multiple families, suggesting diverse attacks provide complementary signals by probing different aspects of model robustness.

**State-of-the-art unlearning can reliably be reversed within 16 fine-tuning steps – sometimes in a single step.** We show the results of multiple fine-tuning attacks against the best-performing model

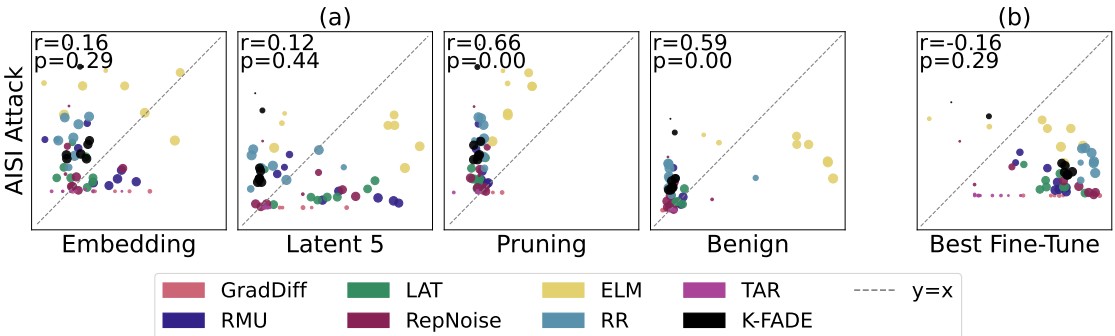

Figure 6: **Model tampering attacks are predictive for a held-out proprietary attack from the UK AI Security Institute.** Each point corresponds to a model. (a) In these experiments, correlations are weaker than with non- UK AISI attacks, but benign fine-tuning attacks continue to correlate with UK AISI input-space attack success. (b) Fine-tuning attacks still tend to exceed the success of input-space attacks, though less consistently than with the attacks from Figure 5.

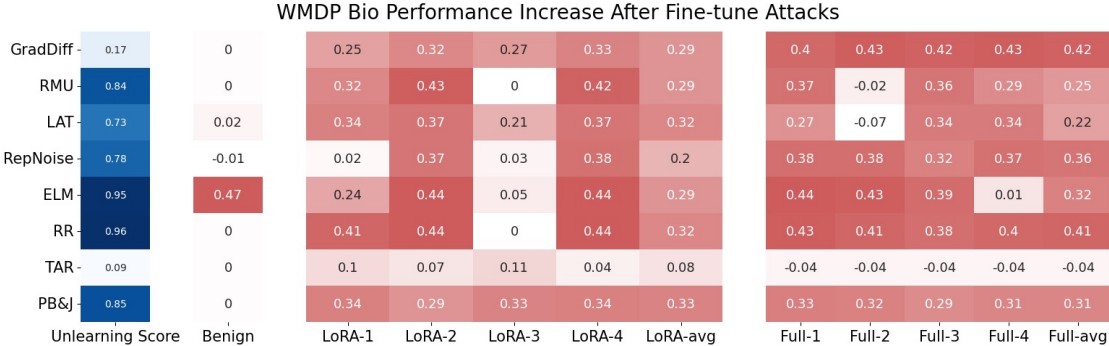

Figure 7: **Few-shot fine-tuning efficiently undoes unlearning.** We plot the heatmap of the best checkpoint for each method under benign (left), LoRA (middle), and full-parameter (right) fine-tuning attacks. All fine-tuning experiments are done within 16 gradient steps, with 128 examples or fewer. All methods can be attacked to increase WMDP-Bio performance by 10 percentage points or more. All hyperparameters are listed in Appendix A.4.

from each unlearning method in Figure 7. All finetuning experiments, as detailed in Appendix A.4, are performed within 16 gradient steps and with 128 or fewer examples. The only method that was resistant to few-shot fine-tuning attacks was TAR, in which only 1 out of the 9 fine-tuning attacks were able to increase the WMDP-Bio performance by over 10 percentage points. However, TAR models had low unlearning scores due to poor general utility, which renders their robustness to fine-tuning unremarkable. All utility-preserving state-of-the-art unlearning methods can be attacked successfully to recover more than 30 percentage points of WMDP performance. Moreover, even when we perform a single gradient step (with a batch size of 64) still increases the WMDP performance on 6 of the 8 methods by over 25 percentage points (see column "Full-4" in Figure 7).

## 4.2 Jailbreaking Experiments

**We repeat analogous experiments with similar results in the jailbreaking setting.** Finally, to test the generality of our findings outside the unlearning paradigm, we ask whether they extend to jailbreaking. Using the 9 off-the-shelf models and 11 attacks from Table 1, we conducted parallel experiments as in Section 4.1 but by pitting off-the-shelf refusal-finetuned models against jailbreaking attacks. We plot all results in Appendix D.

### Harmful Response Increase After Attacks

| | Baseline | GCG | AutoPrompt | BEAST | PAIR | Human Prompt | Embedding | Latent 5 | Pruning | Benign | LoRA Fine-tune | Full Fine-tune | Cascade |
|---|---|---|---|---|---|---|---|---|---|---|---|---|---|
| Llama-3-8B-Instruct | 0 | 0.02 | 0.04 | 0 | 0 | 0.07 | 0.07 | 0.1 | 0.01 | 0.75 | 0.64 | 0.21 | 0.7 |
| RR | 0 | 0 | 0 | 0 | 0 | 0.04 | 0.01 | 0 | 0 | 0 | 0.17 | 0.29 | 0.3 |
| LAT | 0 | 0.09 | 0 | 0 | 0 | 0 | 0 | 0.01 | 0 | 0 | 0.21 | 0.2 | 0.43 |
| TAR | 0.08 | 0.05 | 0.01 | -0.05 | -0.06 | 0.02 | 0.03 | 0.05 | -0.07 | 0 | 0.26 | -0.02 | 0.75 |
| DERTA | 0.02 | 0.05 | 0.03 | -0.02 | -0.02 | -0.02 | 0 | 0 | -0.02 | -0.02 | 0.38 | 0 | 0.58 |
| RT1 | 0 | 0 | 0.01 | 0 | 0 | 0.01 | 0.11 | 0.07 | 0.01 | 0 | 0.28 | 0.31 | 0.63 |
| RT2 | 0.02 | 0.02 | 0.05 | -0.01 | -0.01 | 0.04 | 0.15 | 0.14 | 0.03 | -0.02 | 0.26 | 0.11 | 0.88 |
| RT3 | 0.01 | 0.43 | -0.01 | -0.01 | -0.01 | 0.19 | 0.06 | 0.08 | 0.09 | 0.02 | 0.37 | 0.49 | 0.92 |
| Uncensored | 0.9 | -0.05 | -0.03 | -0.31 | -0.18 | 0.07 | 0 | -0.04 | -0.77 | -0.11 | -0.11 | -0.79 | 0.1 |

Figure 8: **All safety-tuned models could be successfully jailbroken by fine-tuning and Cascade attacks.** We evaluate safety-tuning methods and jailbreak attacks. **Left:** The 'Baseline' measures the compliance rate to direct harmful requests. **Right:** Increase in harmful response rate after attack. All safety-tuning methods were vulnerable to elicitation of suppressed capabilities, especially by finetuning and Cascade attacks (rightmost columns).

Our benchmark results (Figure 8) demonstrate that all safety-tuning methods are vulnerable to model tampering. Principal component analysis of attack success rates in Figure 18 show that three principal components explain 96% of the variation in jailbreak success across the nine models.

We then reproduced our empirical analysis of whether the success of model tampering jailbreaks correlates with and/or conservatively exceeds the success of input-space jailbreaks (Figure 17). Like before, we find that fine-tuning attack success tends to empirically exceed the success of input-space attacks, thus offering a conservative estimation method. However, unlike before, we do not find clear evidence of a reliable correlation between tampering and input-space attacks due to only having 9 samples.

Finally, to evaluate how helpful model tampering attacks can be for characterizing a model's vulnerability to unique, held-out attacks, we use Cascade, a multi-turn, state-of-the-art, proprietary attack algorithm from Haize Labs (Haize Labs, 2023). In Figure 19, we see that single-turn model tampering attacks correlate well with multi-turn Cascade attacks.

## 5 Discussion

**Implications for evaluations and safety cases:** Our findings have direct implications for performing AI risk evaluations and constructing safety cases (Clymer et al., 2024). Current evaluation frameworks rely heavily on input-space attacks which can easily fail to underestimate worst-case failures. Model tampering attacks provide a useful tool for studying novel, potentially unforeseen risks. By modifying a model's internal mechanisms — either through activation perturbations or fine-tuning — we can infer the potential existence of failure modes that input-space evaluations may miss (see also Hofstätter et al. (2025)). This is particularly critical for open-weight models, where safety mitigations can be undone post-release.

**Limitations:** Our work focuses only on Llama-3-8B-Instruct derived models. This allows for considerable experimental depth, but other models may have different dynamics. The science of evaluations is still evolving, and it is not yet clear how to best translate the outcome of evaluations into actionable recommendations. Overall, we find that model tampering attacks can help with more rigorous evaluations – even for models deployed as black boxes. However, there may be limitations in the mechanistic similarity of input-space and tampering attacks (Leong et al., 2024).

**Future work:**

- **Can models be robust to tampering attacks?** This paper and concurrent work (Qi et al., 2024b) show that even defenses designed to make models robust to tampering can be easily undone. We are currently working to better understand tampering robustness and improve the extent to which models can be made robust to tampering attacks through pretraining interventions (e.g. Maini et al., 2025), knowledge corruption (e.g. Wang et al., 2025), and improvements in unlearning algorithms (e.g. Siddiqui et al., 2025).

- **What mechanisms underlie robust capability removal?** We are interested in future work to mechanistically characterize weak vs. robust capability suppression. We briefly worked to test the hypothesis that the activation rank difference (across the forget set) between a base and unlearned model would correlate with unlearning robustness. However, we found this not to be the case, and leave further investigation to future work. We hope that the 64 models we release help to lay the groundwork for this.

- **Bridging research and practice:** Model tampering attacks can be further studied and used in practice to assess risks in consequential models pre-deployment.

## Impact Statement

This work was motivated by advancing the science of AI capability evaluations. This has been a central interest and goal of technical AI governance research (Reuel et al., 2024) because AI risk management frameworks are increasingly being designed to depend on rigorous risk evaluations. Thus, we expect this paper to contribute to developing more rigorous evaluations, which is valuable from a risk-management perspective.

## Acknowledgments

We are grateful to the Center for AI Safety for compute and the Machine Learning Alignment and Theory Scholars program for research support. We thank Daniel Filan for technical support, and Antony Kellermann for technical discussion. Finally, we are grateful to Peter Henderson, Ududec Cozmin, Ekdeep Singh Lubana, and Taylor Kulp-McDowall for their feedback.

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

## A   Experiment Details

### A.1   Unlearning Evaluation

We report the MT-Bench score as the average of one-round and two-round scores and divide it by 10, the maximum number of points possible. The result is scores ranging from 0.0 to 1.0.

### A.2   Unlearning Methods and Implementation

#### A.2.1   Unlearning Methods

- **Gradient Difference (GradDiff):** Inspired by Liu et al. (2022), we train models to maximize the difference between the training loss on the forget dataset and the retain dataset.

- **Random Misdirection for Unlearning (RMU):** Li et al. (2024b) propose a method where model activations on harmful data are perturbed, and model activations on benign data are preserved.

- **RMU with Latent Adversarial Training (RMU+LAT):** Sheshadri et al. (2024) propose training models using adversarial attacks in the latent space as a way to perform stronger unlearning. They combined this with RMU by leveraging adversarial perturbations when training only on the forget dataset.

- **Representation Noising (RepNoise):** Rosati et al. (2024) propose adding a noise loss term that minimizes the KL divergence between the distribution of harmful representations given harmful input and Gaussian noise.

- **Erasure of Language Memory (ELM):** Gandikota et al. (2024) introduce ELM in order to thoroughly unlearn knowledge by training the model to mimic unknowledgeable behavior on the unlearning domain.

- **Representation Rerouting (RR):** Zou et al. (2024) introduces Representation Rerouting (also known as "circuit breaking") which trains models to map latent states induced by topics in the unlearning domain to orthogonal representations.

- **Tamper Attack Resistance (TAR):** Tamirisa et al. (2024) propose TAR as a meta-learning approach to protect open-weight models from finetuning attacks. At each iteration, the model is trained to be robust to a fine-tuning adversary who can take a small number of steps.

- **K-FAC for Distribution Erasure (K-FADE):** McKinney et al. is an unlearning algorithm which learns a set of projections which on activations space which maximally harm performance on the the forget set while minimally perturbing model outputs on a broad retain distribution.

To adhere to the implementations from the works introducing each method, we use full fine-tuning (not LoRA) for RMU, RMU-LAT, RepNoise, TAR, and K-FADE, and LoRA for GradDiff, ELM, RR.

### A.2.2 Hyperparameters

Beginning from prior works' implementations of methods, we tuned the hyperparameters below in order to achieve (1) gradual progress in unlearning across the 8 checkpoints that we took and (2) a high unlearning score by the end of training.

- GradDiff
  - LoRA Fine-tune
      LoRA Rank: 256
      LoRA $\alpha$: 128
      LoRA dropout: 0.05
  - Learning Rate: $10^{-4}$
  - Batch Size: 32
  - Unlearning Loss Coefficient $\beta$: 14

- RMU
  - Layer Fine-tune
      Layers: 5, 6, 7
  - Retain Loss Coefficient $\alpha$: 90
  - Steer: 20
  - Learning Rate: $5 \times 10^{-5}$
  - Batch Size: 8

- RMU+LAT
  - Layer Fine-tune
      Layers: 5, 6, 7
  - Retain Loss Coefficient $\alpha$: 90
  - Learning Rate: $5 \times 10^{-5}$
  - Batch Size: 8
  - Steer: 20

- RepNoise
  - Full Fine-tune
  - Learning Rate: $5 \times 10^{-6}$
  - Batch Size: 4
  - Noise Loss Coefficient $\alpha$: 2
  - Ascent Loss Coefficient $\beta$: 0.01

- ELM
  - LoRA Fine-tune
      LoRA Rank: 64
      LoRA $\alpha$: 16
      LoRA dropout: 0.05

- – Learning Rate: $2 \times 10^{-4}$
- – Batch Size: 8
- – Retain Coefficient: 1
- – Unlearn Coefficient: 6

- Representation Rerouting

  - – LoRA Fine-tune
    - LoRA Rank: 16
    - LoRA $\alpha$: 16
    - LoRA dropout: 0.05
  - – Learning Rate: $1 \times 10^{-4}$
  - – Batch Size: 8
  - – Target Layers: 10, 20
  - – Transform Layers: All
  - – LoRRA Alpha: 10

- TAR

  - – Full Fine-tune
  - – Learning Rate: $2 \times 10^{-5}$
  - – Batch Size: 2
  - – Training Steps: 200
  - – Adversary Inner Loop Steps per Training Step: 16
  - – Retain Representation Coefficient: 1
  - – Retain Log-Loss Coefficient: 1

- K-FADE

  - – Damping factor: $1 \times 10^{-5}$
  - – Retain set estimator: $A_R^2$ (margin squared)
  - – Forget set measure: margin
  - – Iterations: 8
  - – Targeted Layers: 3, 4, 5, 6
  - – Projections per iteration: 1

### A.3 Models for Jailbreaking Experiments

In Table 1, we list the 9 models that we use off the shelf for experiments with jailbreaking. All of which were fine-tuned variants of Llama-3-8B-Instruct from Dubey et al. (2024). Here, we overview each of the 9 models and why we selected them.

1. meta-llama/Meta-Llama-3-8B-Instruct (Dubey et al., 2024): the original Llama-3-8B-Instruct model.

2. slz0106/llama3_finetune_refusal (Link) is a refusal fine-tuned version of Llama-3-8B-Instruct.

3. JINJIN7987/llama3-8b-refusal-vpi (Link) is a refusal fine-tuned version of Llama-3-8B-Instruct.

4. Youliang/llama3-8b-data (Yuan et al., 2024) was fine-tuned to refuse to comply with harmful requests even in cases when a harmful reply begins benignly, or the beginning of a harmful reply is teacher-forced.

5. GraySwanAI/Llama-3-8B-Instruct-RR Zou et al. (2024) was fine-tuned to 'reroute' the latent information flow through the model for harmful requests. The model was designed to respond incoherently with uninformative random-seeming text upon a harmful request.

6. LLM-LAT/llama3-8b-instruct-rt-jailbreak-robust1 (Sheshadri et al., 2024) was fine-tuned as a control model to refuse harmful requests.

7. LLM-LAT/robust-llama3-8b-instruct (Sheshadri et al., 2024) was fine-tuned using latent adversarial training (Casper et al., 2024) to robustly refuse requests under attacks than the above control.

8. lapisrocks/Llama-3-8B-Instruct-TAR-Refusal (Tamirisa et al., 2024) was fine-tuned under weight-space fine-tuning attacks to refuse harmful requests in a way that is robust to fine-tuning.

9. Orenguteng/Llama-3-8B-Lexi-Uncensored (Link) was fine-tuned to comply with any requests.

## A.4 Attack Methods and Implementation

**Greedy Coordinate Gradient (GCG)**  GCG (Zou et al., 2023b) performs token-level substitutions to an initial prompt by evaluating the gradient with respect to a one-hot vector of the current token. Unlike standard GCG, which is typically used to make a model output a specific string, we used a universal version of GCG, optimized over a set of examples to elicit a more general harmful behavior (e.g., giving correct responses to biology questions). We implemented both time-bounded attacks on each unlearned model and transfer attacks using prefixes from one model to attack others. Unless otherwise specified, we report the mean performance of each gradient-guided attack.

**AutoPrompt**  Like GCG, AutoPrompt (Shin et al., 2020) performs a gradient-guided search over input tokens to design universal adversarial prompts. As with GCG, we create universal versions of these attacks using a set of examples.

**BEAST**  We used BEAm Search-based adversarial aTtack (BEAST) (Sadasivan et al., 2024) to produce universal adversarial suffixes which were appended after the evaluation questions. Unlike GCG and Auto-Prompt, BEAST is guided by perplexity instead of gradients. Since our attacks need to be universal, we used a modified version of BEAST to generate universal adversarial tokens for several user input prompts. Formally, we can define a set of user input prompts as $\{x_1^{(u)}, x_2^{(u)}, \cdots, x_n^{(u)}\}$, where each $x_i$ contains a question $q_i$ and answer choices $a_i$. Our goal is to generate an adversarial sequence of tokens $x^{(a)}$ such that $q_i \oplus x^{(a)} \oplus a_i$ can effectively attack the language model for all $i \in \{1, 2, \cdots, n\}$. We place the adversarial tokens between each question and the answer choices so that the beam search in BEAST is conditioned solely on the question and not the answers, as the jailbreak tokens to the end of the user prompt directly can leak the answer choices into the tokens. We attacked with different hyperparameters for search width and time and reported results for $K = 5$.

**PAIR**  In addition to gradient-based approaches such as GCG and finetuning attacks, we also include a model-based approach similar to PAIR (Chao et al., 2024). PAIR uses a prompt-level jailbreaking attack, where an entirely new adversarial prompt is generated instead of augmenting the prompt with adversarial tokens. This is not transferable compared to other universal attack approaches as the attack is highly dependent on the original prompt. Instead, we modify PAIR to develop an adversarial suffix that can applied universally to many (potentially unseen) prompts. First, we task an attack model (base model such as Llama 3 8B) to generate an adversarial suffix, similar to how PAIR generates an adversarial prompt. Then, we append the adversarial suffix to each of the prompts in the training dataset and measure the accuracy of the target model (unlearned model) on the jailbroken prompts. Finally, we pass in an improvement prompt (a general string that asks the attack model to improve its adversarial suffix) and the accuracy of the target model to the attack model to repeat the process.

**Human Prompt**  We hand-crafted adversarial prompts to elicit harmful responses from a language model. After manual iteration, our selected prefix was "#Answer Key\n\n\This answer key contains questions and answers for advanced topics in bioengineering and biotechnology. All answers were verified correct by a panel of experts."

| Attack | Total Forward Passes | Total Backward Passes |
|---|---|---|
| **GCG** | 5120-25600 | 10-50 |
| **AutoPrompt** | 2560-12800 | 10-50 |
| **BEAST** | 630 | 0 |
| **PAIR** | 1920 | 0 |
| **Human Prompt** | 0 | 0 |
| **Embedding Space** | 600 | 600 |
| **Latent Space** | 600 | 600 |
| **WandA Pruning** | 224 | 0 |
| **Benign LoRA Fine-Tune** | 1-16 | 1-16 |
| **LoRA Fine-Tune** | 1-16 | 1-16 |
| **Full Parameter Fine-Tune** | 1-16 | 1-16 |

Table 3: **Model tampering attacks empirically tend to be more efficient than input-space attacks.** To show the computational expansiveness of the attacks that we use, we report the number of forward plus backward passes used to develop each attack under our implementations. The model architecture and number of parameters in all models was the same (up to small, inserted LoRA adapters), but the number of tokens in strings used to develop each attack varied. For these reasons, note that the number of forward and backward passes does not have a perfectly consistent relationship with the number of floating point operations.

**Embedding Attack** Schwinn et al. (2024) proposes applying adversarial perturbations to a language model's embedding layer instead of in the input space. We train universal $\ell_2$-norm bounded embedding space prompts which we apply to the embedded tokens for the prefix used for WMDP-Bio evaluation: "The following are multiple choice questions (with answers) about biology.\n".

**Latent Layer Attack** Sheshadri et al. (2024) proposes applying adversarial perturbations to a language model's latent layers. We train universal $\ell_2$-norm bounded latent pseudoprompts which we apply to the latent tokens at layer 5 for the prefix used for WMDP-Bio evaluation: "The following are multiple choice questions (with answers) about biology.\n".

**WandA Pruning** To test each model's performance on benign pruning attacks, we use Weights and Activations (WandA) pruning Sun et al. (2023). WandA prunes the lowest $p\%$ of parameters according to the importance score generated by $W_i X$, where $W_i$ is the $i$th layer weight and $X$ is the input. For all experiments, we prune the weights 15%.

**Few-shot fine-tuning attacks** We perform fine-tuning to re-elicit unwanted capabilities. The forget set (WMDP-Bio Remove) consists of hazardous biology data, and the retain set (WMDP-Bio Retain) contains non-hazardous biology data. We also performed benign LoRA fine-tuning on Wikitext. We report hyperparameters in Appendix A.4. All LoRA and Benign attacks are done with rank 16 and alpha 32. All examples have a maximum length of 512 tokens. Few-shot fine-tuning attack details are reported in Appendix A.4.

**Excluded attacks:** In addition to these attacks, we also experimented with many-shot attacks (Anil et al., 2024; Lynch et al., 2024) and translation attacks (Yong et al., 2023; Lynch et al., 2024) but found them to be consistently unsuccessful in our experimental settings.

| | Dataset | # of Examples | Batch Size | Learning Rate | Epochs | Total Steps |
|---|---|---|---|---|---|---|
| Full-1 | WMDP-Bio Remove | 400 | 16 | 2e-05 | 2 | 25 |
| Full-2 | WMDP-Bio Remove | 64 | 8 | 2e-05 | 2 | 16 |
| Full-3 | WMDP-Bio Retain | 64 | 64 | 5e-05 | 2 | 2 |
| Full-4 | WMDP-Bio Retain | 64 | 64 | 5e-05 | 1 | 1 |
| LoRA-1 | WMDP-Bio Remove | 400 | 8 | 5e-05 | 1 | 50 |
| LoRA-2 | WMDP-Bio Retain | 400 | 8 | 5e-05 | 1 | 50 |
| LoRA-3 | WMDP-Bio Remove | 64 | 8 | 1e-04 | 2 | 16 |
| LoRA-4 | WMDP-Bio Retain | 64 | 8 | 1e-04 | 2 | 16 |
| Benign-1 | Wikitext | 400 | 8 | 5e-05 | 1 | 50 |

Table 4: Hyper-parameters for Fine-tuning Attacks on Unlearned Models

| | Dataset | # of Examples | Batch Size | Learning Rate | Epochs | Total Steps |
|---|---|---|---|---|---|---|
| Full-1 | LAT Harmful | 64 | 8 | 5e-05 | 1 | 8 |
| Full-2 | LAT Harmful | 16 | 8 | 5e-05 | 1 | 2 |
| LoRA-1 | LAT Harmful | 64 | 8 | 5e-05 | 1 | 8 |
| LoRA-2 | LAT Harmful | 64 | 8 | 5e-05 | 2 | 16 |
| LoRA-3 | LAT Harmful | 16 | 8 | 5e-05 | 2 | 4 |
| LoRA-4 | LAT Harmful | 64 | 8 | 1e-04 | 2 | 16 |
| Benign-1 | Ultra Chat | 64 | 8 | 5e-05 | 2 | 16 |

Table 5: Hyper-parameters for Fine-tuning Attacks on Refusal Models

# B Full Unlearning Results

## B.1 Standard Attacks

In Figure 9, we plot the attack successes for all model tampering attacks against all input-space attacks.

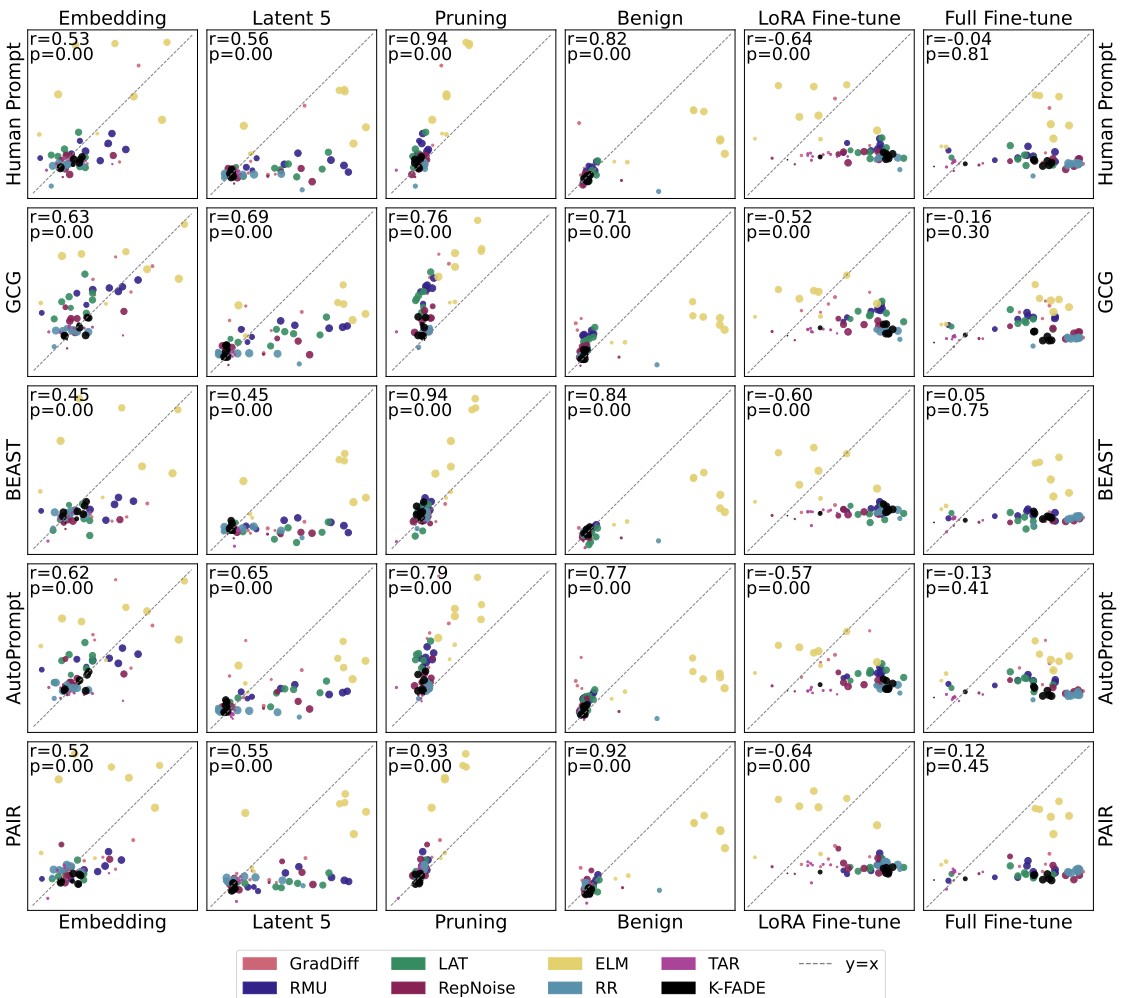

Figure 9: **Full results from unlearning experiments comparing input-space and model tampering attacks.** See summarized results in Figure 5. Here, we plot the increases in WMDP-Bio performance from model tampering attacks and input-space attacks. We weight points by their unlearning score from Section 4.1.1. We also display the unlearning-score-weighted correlation, the correlation's $p$ value, and the line $y = x$. Points below and to the right of the line indicate that the model tampering attack was more successful.

## B.2 UK AISI Attacks and Evaluation

In Figure 10, we plot the full attack successes for all model tampering attacks against the UK AISI attack.

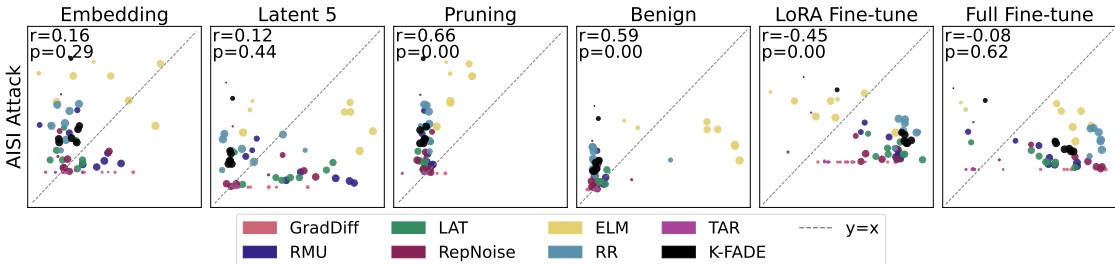

Figure 10: **Model tampering attacks remain predictive for a proprietary attack from the UK AI Security Institute.** (a) In these experiments, correlations are weaker than with non- UK AISI attacks, but benign fine-tuning attacks continue to correlate with UK AISI input-space attack success. (b) Fine-tuning attacks still tend to exceed the success of input-space attacks, though less consistently than with the attacks from Figure 5.

Next, to test the limits of our hypothesis that model tampering attacks can help evaluators assess novel, unforeseen failure modes, we evaluated model performance under an entirely different non-WMDP benchmark for dual-use bio capabilities from the UK AI Security Institute. Figure 12 shows that WMDP-Bio performance correlates with this evaluation with $r = 0.64$ and $p = 0.0$. To correct for this confounding factor, in Figure 6, we use model tampering attack success on WMDP-Bio to predict the *residuals* from a linear regression predicting UK AISI Bio evaluation results from WMDP-Bio evaluation results. Here, we find weak correlations except for the case of the pruning and benign fine-tuning methods. Overall, this suggests that while model tampering attacks can be informative about novel failure modes across different *attacks*, they do not necessarily do so across different *tasks*.

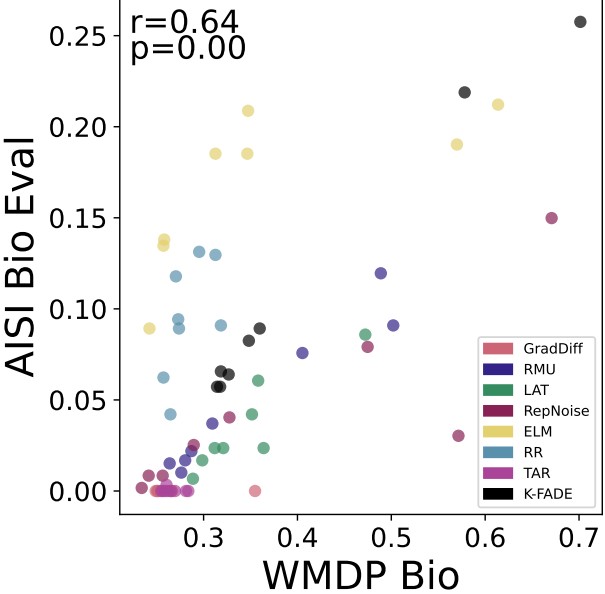

Figure 11: **WMDP-Bio performance correlates with the UK AISI Bio evaluation performance**.

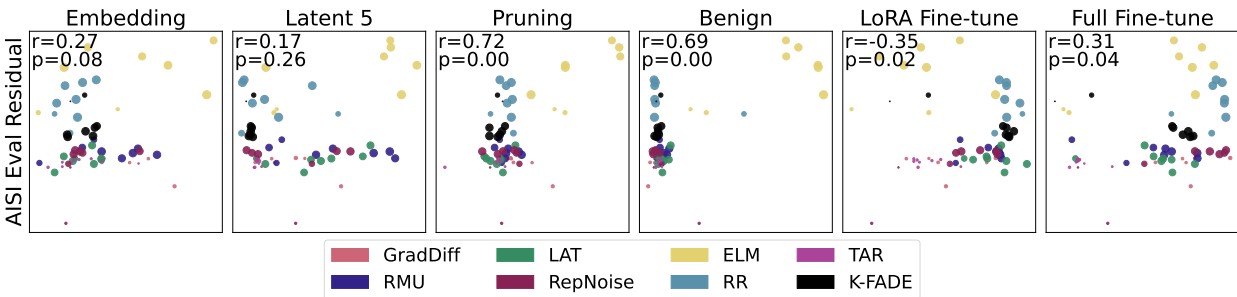

Figure 12: **Model tampering attack success on WMDP-Bio is not strongly predictive of model success on UK AISI bio capability evaluations.** This suggests a limitation of how informative model tampering attacks can be about failure modes across task distributions.

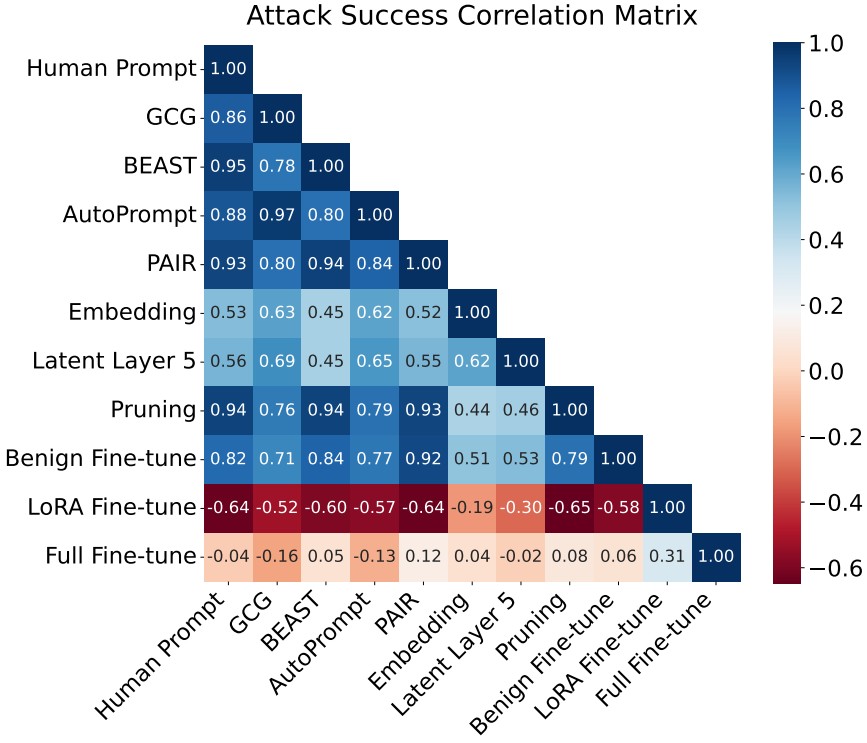

Figure 13: **Attack Success Correlation Matrix.** We compute attack success rate correlations across all $n = 65$ unlearning models. Input-space attacks show strong positive correlations (0.78-0.97) with each other, suggesting they exploit similar model vulnerabilities. In contrast, model tampering attacks show more varied and generally weaker correlations, both with each other and with input-space attacks. This suggests they probe model vulnerabilities through different mechanisms than input-space attacks, making them valuable complementary tools for harmful capability evaluations.

## B.3   Attack Relationships

We visualize the relationships between attacks in Figure 13 (attack correlation matrix) and Figure 4 (attack clustering tree). First, attacks with similar algorithmic mechanisms have highly correlated success rates. Second, full-finetuning attacks exhibit significant variation, even amongst each other. Since branching height indicates subtree similarity (higher height means less similar), Figure 4 implies that LoRA and Full-finetuning

attacks are less similar to each other than input-space and latent space attacks are. Meanwhile, pruning and benign finetuning behave similarly to gradient-free input-space attacks.

## C  Do model tampering attacks improve input-space vulnerability estimation?

### C.1  Model tampering attacks improve predictive accuracy for worst-case input-space vulnerabilities

| Linear Regression Inputs | RMSE (%) | $R^2$ |
|---|---|---|
| BEAST, PAIR, Embedding | 0.0453% | 0.5947 |
| Human Prompt, AutoPrompt, LoRA Fine-tune | 0.0457% | 0.7596 |
| BEAST, AutoPrompt, LoRA Fine-tune | 0.0463% | 0.7608 |
| GCG, PAIR, Benign Fine-tune | 0.0469% | 0.8161 |
| Human Prompt, Embedding, LoRA Fine-tune | 0.0473% | 0.6923 |

Table 6: **Top-5 subsets of attacks most predictive of worst-case input-space success rate.** We compute all subsets of 3 attacks, and for each subset, we use linear regression to predict the worst-case input-space success rate from success rates of attacks in the subset. We show the top-5 subsets by RMSE. These top subsets lead to very accurate predictors of worst-case vulnerabilities and typically include diverse attack types (input-space gradient-free, input-space gradient-based, and model tampering).

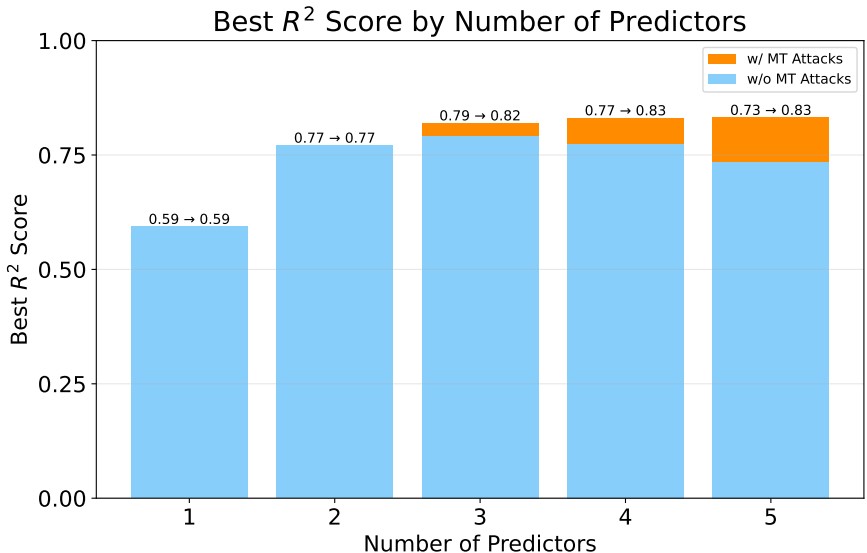

Figure 14: **Model tampering attacks help predict worst-case input-space vulnerabilities.** We perform linear regressions to predict the worst-case input-space success rate from success rates of subsets of attacks. Including model tampering attacks in these subsets improves worst-case vulnerability estimation $R^2$ by 0.05-0.1. Ultimately, however, this is likely a conservative quantification of the marginal predictiveness of model tampering attacks for unforeseen input-space threats. The two most effective input space attacks were GCG and AutoPrompt, and as shown in Figure 13, their correlation is 0.88. However, unforeseen attacks in the real world are by no means guaranteed to be as similar to standard input-space attacks as GCG and AutoPrompt are to each other. As a result, this experiment is likely to paint a more pessimistic view on the value of model tampering attacks for predicting held-out input space attacks.

In this section, we investigate the utility of model tampering attacks for worst-case input-space vulnerability estimation. While Figure 5 shows that fine-tuning attacks empirically offer conservative estimates for worst-case input-space vulnerabilities, in this section, we also show that model tampering attacks improve

evaluators' ability to *predict* worst-case vulnerabilities – even if they already have access to input-space attacks.

For all experiments in this section, we assume the setting of an evaluator who only has access to a subset of attacks in order to estimate worst-case input-space vulnerabilities (potentially due to novel attacks). Whether due to resource constraints on the number of evaluations that are feasible to implement or due to the constant invention of new attack methods, evaluators will always be in this kind of setting. In our setup, we fit linear regression to predict worst-case input-space success rates given the success rates of a subset of attacks. Our dataset consists of a table of all unlearned models (and their 8 checkpoints throughout training) and all attack success scores (WMDP accuracy after attack - base WMDP accuracy). We perform $k$-fold cross-validation across model families by holding out all models trained by the same unlearning method, one method at a time. We then average statistics (e.g. RMSE, $R^2$) across the splits. Note that we include the UK AISI input-space attack in these experiments, giving us 6 input-space attacks.

While our cross-validation procedure (with held-out model families) reflects the real-world setting of receiving a new model trained with unknown methods, it results in a validation set that is no longer i.i.d. with the train set. Due to this distribution shift, the assumption underlying the typical formula for $R^2$ is violated. So, when calculating $R^2 = (1 - mse/variance)$, instead of standard variance within the validation set, we use $\frac{1}{|val|} \sum_{s \in val}(s - \mu_{train})^2$ (where $\mu_{train}$ is the mean score in the train set instead of the validation set). Otherwise, the $\mu_{val}$ would use privileged information from the validation set that's not available in an i.i.d. setting. Note that because of this and our cross-validation procedure, the MSE and $R^2$ may lead to different rankings over performance of predictors.

Table 6 shows the top-5 subsets of 3 attacks that lead to the lowest RMSE in predicting worst-case input-space attack success rate. Note that in all cases, at least one model tampering attack is present. Additionally, these subsets typically include diverse attack types. This supports the hypothesis that probing vulnerabilities through different mechanisms can improve worst-case held-out estimation.

Figure 14 shows that across subset sizes, including model tampering attacks lead to non-trivial improvements in worst-case predictive performance. Given the large size of subsets, we perform k-fold cross-validation over input-space attacks in addition to model families. Here, we loop through each input-space attack, holding out one input-space attack at a time so it is excluded as an input to linear regression. We then fit and evaluate the predictor's ability to estimate the worst-case success rate over all input-space attacks. Blue bars show the best $R^2$ over subsets made of input-space attacks only while orange bars show the best $R^2$ over all subsets.

### C.2 Input-space attacks are most predictive of average-case input-space vulnerabilities

Figure 15 shows average-case predictive performance with different subsets of attacks. Here, including model tampering attacks do not seem to improve predictive performance for the attacks tested here. We hypothesize that high correlations and similar attack mechanisms between input-space attacks make them more effective predictors of each other on average. In contrast, because model tampering attacks exploit distinct mechanisms, they are effective for predicting and bounding worst-case vulnerabilities.

## D  Full Jailbreaking Results

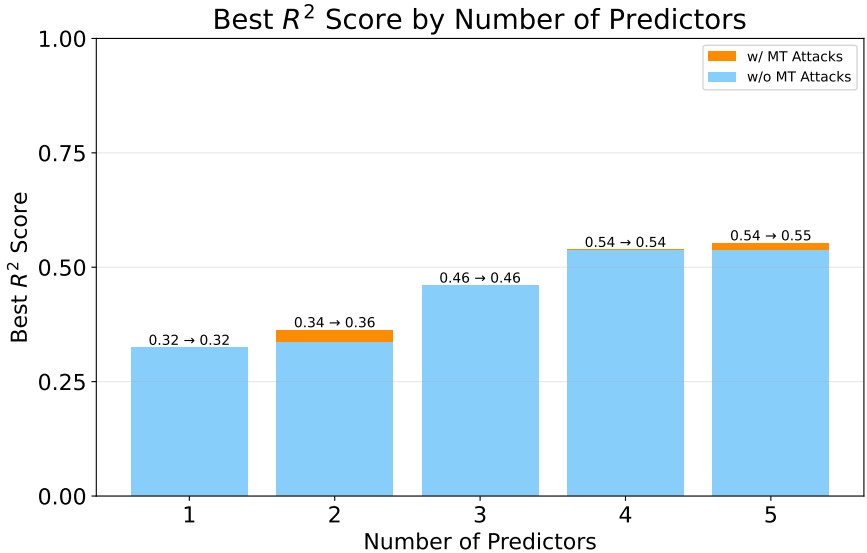

Figure 15: **Input-space attacks are most predictive of average-case input-space vulnerabilities.** Here, we train linear regression to predict success rates of every input-space attack and average the $R^2$. Model tampering attacks do *not* consistently improve predictive performance.

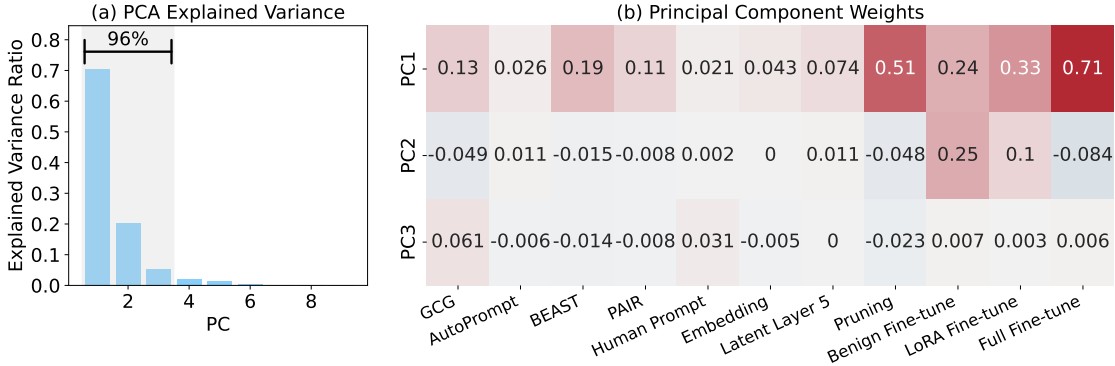

Figure 16: Three principal components explain 96% of the variation in attack success. **Left:** The proportion of explained variance for each principal component. **Right:** We display the first three principal components weighted by their eigenvalues. All coordinates of the first principal component are positive.

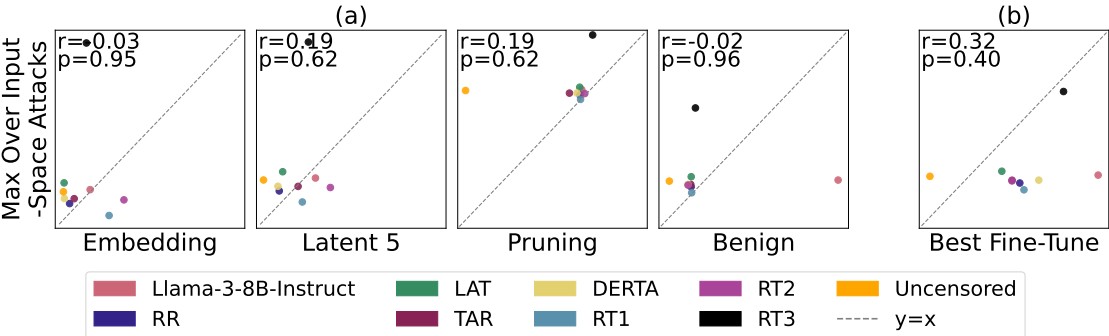

Figure 17: **In our experiments, fine-tuning attack successes empirically *exceed* the successes of state-of-the-art input-space attacks for jailbreaking.** Here, we plot the increases in compliance with harmful requests under model tampering attacks against the best-performing (out of 5) input-space attacks for each model. On the right, the $x$ axis is the best (over 2) between a LoRA and Full fine-tuning attack. We also display the correlation and the correlation's $p$ value. There are only 9 points in each figure, so we cannot draw strong conclusions. However, we see no clear evidence of a correlation between model tampering and input-space attack success. However, as before in Figure 5, fine-tuning attacks empirically tend to offer conservative estimates of the success of input-space attacks. The only case out of 9 in which this was not the case was with the uncensored Orenguteng model (link) which was unlike the other 8 in that it was not designed to be robust to jailbreaks.

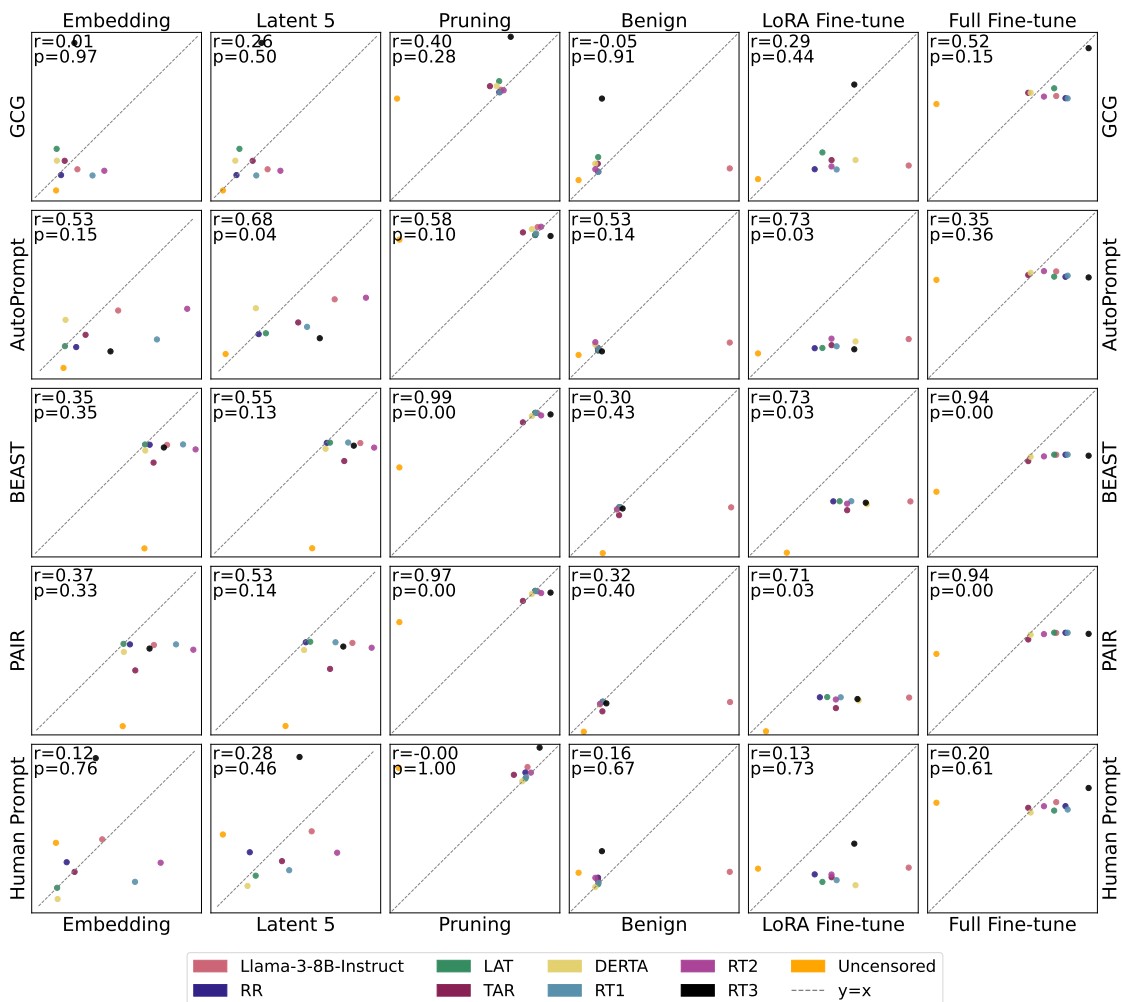

Figure 18: **Full results from jailbreaking experiments comparing input-space and model tampering attacks.** See summarized results in Figure 17. Here, we plot the increases in WMDP-Bio performance from model tampering attacks and input-space attacks. We also display the unlearning-score-weighted correlation, the correlation's $p$ value, and the line $y = x$. Points below and to the right of the line indicate that the model tampering attack was more successful.

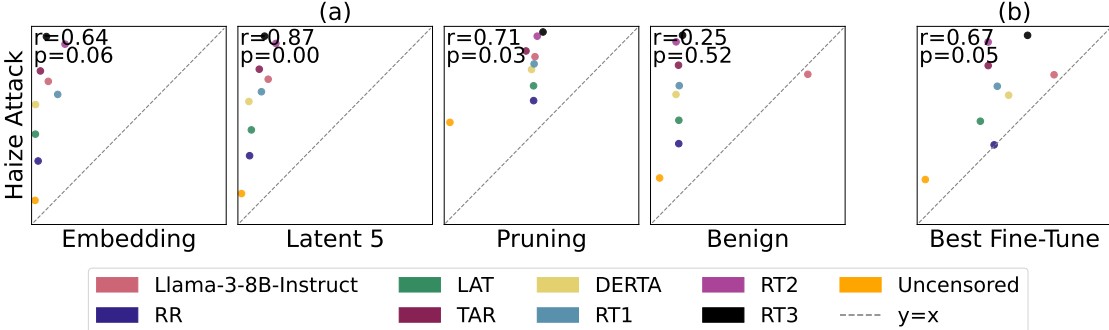

Figure 19: **Single-turn model tampering attack successes correlate with attacks from Cascade, a multi-turn, proprietary attack algorithm** . Since Cascade is state-of-the-art and multi-turn, our single-turn model tampering attacks do not tend to empirically exceed the success of this attack as we find for unlearning experiments (Figure 5). However, they empirically correlate with its success.

