# OpenReview forum: "Model Tampering Attacks Enable More Rigorous Evaluations of LLM Capabilities"
_TMLR — Accepted by TMLR_

### Review · Reviewer_tXtq · 2025-05-05

**Summary Of Contributions:**

This paper proposes model tampering attacks as a complementary approach to input-space evaluations for assessing LLM risks. By modifying model weights or activations, tampering enables more rigorous stress-testing. The authors benchmark unlearning and safety fine-tuning methods, showing that tampering predicts input-space vulnerabilities, and harmful capabilities can be introduced with few fine-tuning steps.

**Audience:**

Yes

**Broader Impact Concerns:**

The paper does not raise significant broader impact concerns.

**Claims And Evidence:**

Yes

**Requested Changes:**

1. The authors should add experiments on additional models, even if only simple validations, to support the generality of their core conclusions beyond LLaMA-3-8B.
2. The paper would benefit from a discussion of potential defense mechanisms against model tampering attacks, which is currently missing.
3. Figures should be placed closer to the relevant textual descriptions to improve clarity and readability.
4. The visual clarity of figures needs improvement; for example, the precision of reported results in Figure 3 should be made consistent.
5. Both the result plots and illustrative diagrams could be made more polished and visually refined to enhance the overall presentation quality.

**Strengths And Weaknesses:**

Strengths:
1. The paper introduces model tampering attacks, demonstrating how these attacks can predict and conservatively estimate the success of input-space attacks. This approach offers a new perspective for evaluating model safety, revealing potential vulnerabilities in traditional input-space attacks.
2. The authors conduct wide experiments across different models, attack strategies, and evaluation metrics. By testing 65 models, the paper provides detailed quantitative results that showcase the performance differences of various attacks and defense strategies.
3. The paper explores the relationships between tampering attacks and input-space attacks. It demonstrates that tampering attacks can predict the success of input-space attacks and help assess worst-case performance.

Weakness:
1. The paper primarily focuses on experiments conducted with the Llama-3-8B model, which may not fully capture the behavior of other models. The reliance on a single model restricts the generalizability of the results and raises concerns about whether the findings are applicable across a wider range of models.
2. While the paper identifies correlations between model tampering and input-space attacks, it does not provide a comprehensive analysis or explanation of why these two types of attacks exhibit such relationships. The lack of in-depth mechanistic insight leaves an open question regarding the underlying factors that link model tampering with input-space attack success.
3. The paper discusses the challenges of tampering attacks and the difficulty of achieving robustness against them, but it does not provide specific defense methods or approaches to enhance model robustness against tampering. This limits the practical applicability of the findings for mitigating these vulnerabilities.

---

> ### Author Response · Authors · 2025-05-25
> **Thanks + reply**
>
> We are thankful for your time and help, especially related to discussing defenses and explanations for what makes models more or less tamper-resistant. We were glad to hear that you found this work to be comprehensive and novel! See responses to key points below.
>
> ## 1. Models
>
> > The paper primarily focuses on experiments conducted with the Llama-3-8B model
>
> **We agree:** We acknowledge that only working with Llama3-8B-derived models is a limitation, and we agree with the value of continued research on tampering attacks in other model families.
>
> **In defense of our approach – lots of experimental depth:** While we only worked with Llama3-8B-derived models, this choice helped us to have a high degree of experimental depth, working with 2 tasks (bio unlearning & jailbreak robustness), 8 capability suppression methods, 11 attacks, and a total of 74 models. Focusing on depth allowed us to have more rigorous insights and release our suite of 64 models. We would argue that this paper was reasonably scoped in that it took our team concerted effort over ~6 months to implement, tune, and run all of the attacks and defenses that we used. Duplicating our experiments with other model families would be a challenge because hypers do not generally transfer well across model families. We would rather produce work that experiments thoroughly and rigorously with one model family than perfunctorily with multiple. **In your final evaluation, please weigh our investigation into a single model family against the high degree of algorithmic diversity in our experiments.** Overall, we would argue that the volume of experimentation and insights is more than enough to usefully contribute and meet a reasonable standard. Valuable papers often only experiment with a single model family. For example, reviewer J9HP mentioned [Wei et al. (2024)](https://dl.acm.org/doi/10.5555/3692070.3694226) who only experiment with Llama2 models and were published at ICML. Unlike our paper, [Wei et al. (2024)](https://dl.acm.org/doi/10.5555/3692070.3694226) only experiment with one task  – safety (via AdvBench) – while we experiment with two – biology unlearning and jailbreak robustness. We also note that they only experiment with the base models – not any unlearning or refusal posttraining methods.
>
> **In followup work, we are working with OLMo-2 and SmolLM models.** In our current work, we are not exactly replicating what we did for this paper, but it is focused on substantially improving the state-of-the-art for tamper robustness.
>
> **Question 1.1: What do you think about how breadth weighs against depth?**
>
> ## 2. Mechanistic explanation
>
> > it does not provide a comprehensive analysis or explanation of why these two types of attacks exhibit such relationships. The lack of in-depth mechanistic insight
>
> **We ran some experiments to study this and did not find an explanation.** We agree with studying this. While we were working on the paper, we spent ~2 weeks on testing the hypothesis that the *activation rank difference* between the base model and a fine-tuned model on the unwanted data distribution (e.g. bio text) would correlate with the fine-tuned model’s tamper resistance. Unfortunately, we found that activation rank difference could vary dramatically with little chance in tamper resistance.
>
> **Action 2.1: We are working on this, planning to add it to the paper.** We agree and remain interested in this. In response, we are kicking off experiments to quantify the rank difference between the original model and each of our 64 unlearned models on bio data. Based on prior results, we think that there will not be a clear relationship between rank difference and unlearning robustness, but even a negative result may be useful here.
>
> **Question 2.2: Do you have any ideas about what kinds of mechanistic qualities of networks might causally relate to tamper resistance?** We would be curious if you have any hypotheses.

---

> > ### Author Response · Authors · 2025-05-25
> > **Reply continued**
> >
> > ## 3. Defenses
> >
> > > it does not provide specific defense methods or approaches to enhance model robustness against tampering.
> >
> > **Most papers only focus on attacks or defenses.**  We agree with the importance of studying attacks and defenses. We think that this criticism applies to the large majority of research on attacks/defenses for LLMs. We also know that good benchmarking is a prerequisite to making progress on both adversary evaluation and adverse robustness.
> >
> > **We are currently working on tamper resistance defenses.** Followup work of ours is focused on substantially improving the state-of-the-art for tamper robustness. We are currently working to study defenses in depth against tampering which include (1) filtering the models pre-training data (2)  training the model on incorrect information about the unwanted knowledge domain, and (3) using combinations of state-of-the-art unlearning techniques.
> >
> > **Question 3.1: Do you have any ideas or thoughts on improving tamper robustness?**  if you do, we'd be interested since we're currently working on it.
> >
> > > The paper would benefit from a discussion of potential defense mechanisms against model tampering attacks, which is currently missing.
> >
> > **Action 3.2: We added this.** We have a lot of thoughts on this because we are working on it. We added some discussion to section 5 about how it might be possible to improve over current limitations in defenses.
> >
> > ## 4. Figures
> >
> > > Figures should be placed closer to the relevant textual descriptions to improve clarity and readability.
> >
> > **Action 4.1: We updated figure placement.** Thanks for mentioning this.
> >
> > > the precision of reported results in Figure 3 should be made consistent.
> >
> > **Question 4.2: Clarification.** Could you elaborate a little bit on what you mean by “precision” in this context?
> >
> > > result plots and illustrative diagrams could be made more polished
> >
> > **Question 4.3: Clarification.** Could you elaborate on how to polish things?
> >
> > ## What do you think?
> >
> > **Thank you again for taking the time to review the paper and providing helpful feedback! Do the above actions address your concerns with the paper? If not, what further clarification or modifications could we make to improve it?**

---

> > > ### Comment · Reviewer_tXtq · 2025-05-29
> > >
> > > I appreciate the authors' efforts in addressing the concerns raised in my review.
> > >
> > > Regarding the use of a single model family in the experiments, I agree with the authors’ response for prioritizing experimental depth over breadth.
> > >
> > > For the remaining concerns, I have the following comments:
> > >
> > > 1. **Mechanistic explanation and discussion.** I understand the challenges of establishing concrete mechanistic explanations for tamper resistance. I believe the authors are not required to resolve these questions fully. However, it would strengthen the paper to explicitly discuss these open challenges, including negative results, hypotheses tested, and possible future directions. Including such content would help guide future research.
> > >
> > > 2. **Discussion of defenses.** I appreciate the authors’ acknowledgment of this limitation and the addition of some thoughts in Section 5. While detailed defenses may be beyond this paper’s immediate scope, briefly outlining potential strategies, as done in the rebuttal, enhances the paper’s contribution.
> > >
> > > 3. **Figure clarity and presentation.** I respect the authors’ writing and formatting preferences. My comments on figure precision and visual polish were suggestions for improving presentation quality. I encourage the authors to refer to well-presented papers in top venues for inspiration. Since this is not a primary concern in my review, I will not offer specific suggestions here.
> > >
> > > 4. **Revision Highlighting.** To facilitate checking during the final review process, I recommend clearly highlighting the revised content in the paper (e.g., colored highlights). This will help reviewers quickly verify the updates.

---

> > > > ### Author Response · Authors · 2025-05-29
> > > > **Thanks + reply**
> > > >
> > > > Thank you for the feedback and thoughts. We appreciate your time and help.
> > > >
> > > > 1. **Experiments in progress:** We appreciate the thoughts, and we agree with the value of the negative result. We are committed to adding this. So far, we have set up new, fresh experiments for our rank-diff analysis. We should be able to start running the experiment today or tomorrow. Then we will plan, by default, to add these to the appendix.
> > > > 2. **Paper updated:** We have updated the discussion to elaborate on directions that we are pursuing and which we think are promising for improving tamper resistance: pretraining interventions (e.g. [Maini et al. (2025)](https://arxiv.org/abs/2504.16980)), knowledge corruption (e.g. [Wang et al. (2025)](https://alignment.anthropic.com/2025/modifying-beliefs-via-sdf/)), and modifications to unlearning algorithms (e.g. [Siddiqi et al., (2025)](https://arxiv.org/abs/2505.22310v1)).
> > > > 3. **Investigating this:** Thank you. We think there is room for improvement. We ourselves don't have many specific ideas because maybe we have been staring at these figures too long. However, we will ask a few lab members later today for their thoughts on the clarity and aesthetics of the figures overall. We'll work to act on their feedback.
> > > > 4.  **Highlighted version updated:** In the updated version of the paper, post-rebuttal key changes are in pink. Thanks!

---

### Review · Reviewer_rXbX · 2025-05-08

**Summary Of Contributions:**

This paper benchmarks the safety of LLMs under model tampering attacks, particularly focusing on attacks following unlearning interventions. The authors evaluate 9 safety-tuned and 65 unlearned models against 11 capability elicitation attacks (5 input-space, 6 tampering). Key contributions include the discovery that model robustness to these attacks lies in a low-dimensional subspace, and that model tampering attacks can predict or conservatively estimate the success of held-out input-space attacks. The paper also demonstrates that current unlearning methods can be undone with a small number of fine-tuning steps.

**Audience:**

Yes

**Claims And Evidence:**

Yes

**Requested Changes:**

I suggest that the author improve the paper in the following aspects:

- Provide more diverse datasets or include discussion on how results might generalize beyond WMDP-Bio.
- Deepen the analysis of why certain attacks correlate (e.g., the mechanisms behind the PCA clusters).
- Clarify early in the paper that the focus is primarily on unlearning and refusal tuning, and why such selection is enough to represent existing defense scopes.


I’m also curious if it’s valid to experiment with LLMs given different training checkpoints. I suspect this will cause the homogeneous results observed in the paper. For example, if the model tampering attacks still show high correlations with input-space attacks when we compare different models, instead of the same model with different checkpoints from the unlearning process.

**Strengths And Weaknesses:**

**Strengths**

- *Comprehensive Evaluation*. The paper provides a comprehensive evaluation across multiple safety fine-tuned (unlearning) LLMs with 11 capability elicitation attacks. The scope comprehensively covers the existing attack methodologies.

- *Interesting Findings*. The paper shows interesting observations that model tampering attacks related to input space attacks share similarities. The observation can benefit future research in evaluating LLM safety more rigorously.

**Weaknesses**

- *Lack of an evaluated dataset*. The experimental results are mainly derived from the WMDP-Bio dataset. The paper can further benefit from evaluating on more datasets to show the generality of the findings.
- *Lack of in-depth insights*. While the paper identifies empirical correlations (e.g., PCA showing 89% variance explained by 3 components), it lacks in-depth interpretation of these components and the causal mechanisms behind observed clustering in Figure 4. Further analysis could strengthen claims about the robustness subspace and attack similarity.
- *The conclusion mainly focuses on unlearning defense*. The work mainly focuses on unlearning as a defense mechanism. While the paper does include fine-tuned refusal models in the jailbreak setting, expanding defense comparisons beyond unlearning and refusal tuning could broaden impact and applicability.

---

> ### Author Response · Authors · 2025-05-25
> **Thanks + reply**
>
> We are thankful for your time and help, especially related to discussing explanations for what makes models more or less tamper-resistant. We were glad to hear that you found this work to be comprehensive and interesting! See responses to key points below.
>
> ## 1. Tasks
>
> > The experimental results are mainly derived from the WMDP-Bio dataset. The paper can further benefit from evaluating on more datasets to show the generality of the findings…Provide more diverse datasets or include discussion on how results might generalize beyond WMDP-Bio.
>
> **Don’t forget about the jailbreaking experiments.** Our focus is not just on unlearning dual-use biology knowledge. Section 4.2 is entirely about experiments with jailbreak robustness. The task, models, data, and algorithms used in experiments in 4.2 are different than in 4.1.
>
> **Appendix B.2 focuses on proprietary bio-capability evaluations – not just WMDP-bio.** It was appendixed for space, but our experiments in B.2 test the limits of our hypothesis that model tampering attacks can help evaluators assess novel, unforeseen failure modes. We focus on testing whether tampering attacks on one bio task can be marginally predictive of performance on another bio task.
>
> **See also our response No. 1 to J9HP above.**
>
> **Question 1.1: What do you think of the jailbreak experiments from 4.2?** Do you think we should work to present 4.2 more centrally in the paper?
>
> ## 2. Other defenses
>
> > The conclusion mainly focuses on unlearning defense…While the paper does include fine-tuned refusal models in the jailbreak setting, expanding defense comparisons beyond unlearning and refusal tuning could broaden impact and applicability…expanding defense comparisons beyond unlearning and refusal tuning could broaden impact and applicability.
>
> **Unlearning is very general:** As you mention, we study refusal and unlearning techniques. We think that this covers the space of defenses well because they are very general. In our case, “unlearning technique” just refers to a method that aims to suppress specific unwanted capabilities in a model using examples of that harmful behavior. We are not sure what types of techniques this fails to capture and what other classes of techniques would be useful to add. Perhaps this comment refers to scaffolding-based techniques such as training safety classifiers for model inputs/outputs. But model tampering attacks could be used to red-team these types of classifiers as well.
>
> **In defense of our approach – 8 unlearning methods and 9 refusal-trained models is a relatively large number of defenses to study.** Most papers do not work this extensively with capability suppression methods. For example, reviewer J9HP discussed [Wei et al. (2024)](https://dl.acm.org/doi/10.5555/3692070.3694226) (accepted to ICML 2024) who only study base models.
>
> **Question 2.1: Specifically, what should we add?** Understanding what harmful capability suppressiong techniques we may be missing beyond refusal training and unlearning tools would help us understand how to address this.
>
> > Clarify early in the paper that the focus is primarily on unlearning and refusal tuning, and why such selection is enough to represent existing defense scopes.
>
> **Action 2.2: adding this.** Thank you – we agree this is good to clarify. We updated the methods section of the paper to say that we focus on capability suppression tools by studying refusal and unlearning.
>
> ## 3. Studying causal mechanisms
>
> > It lacks in-depth interpretation of these components and the causal mechanisms behind observed clustering in Figure 4.
>
> **We ran some experiments to study this and did not find an explanation.** We agree with studying this. While we were working on the paper, we spent ~2 weeks on testing the hypothesis that the *activation rank difference* between the base model and a fine-tuned model on the unwanted data distribution (e.g. bio text) would correlate with the fine-tuned model’s tamper resistance. Unfortunately, we found that activation rank difference could vary dramatically with little change in tamper resistance.
>
> **Action 3.1: We are working on this, planning to add it to the paper.** We agree and remain interested in this. In response, we are kicking off experiments to quantify the rank difference between the original model and each of our 64 unlearned models on bio data. Based on prior results, we think that there will not be a clear relationship between rank difference and unlearning robustness, but even a negative result may be useful here.
>
> **Question 3.2: Do you have any other ideas about what kinds of mechanistic qualities of networks might causally relate to tamper resistance?** We would be curious if you have any hypotheses.

---

> > ### Author Response · Authors · 2025-05-25
> > **Reply continued**
> >
> > ## 4. Studying checkpoints from the same runs
> >
> > > I’m also curious if it’s valid to experiment with LLMs given different training checkpoints.
> >
> > **We discuss this limitation.** In footnote 4 (Section 4.1.2), we say “Points are not independent or identically distributed, so we only use this “p” value for geometric intuition, and we do not attach it to any formal hypothesis test.” We also note that this limitation does not apply to any of our benchmarking analysis as we base it on the best checkpoint from each run.
> >
> > **Our analysis in Appendix C addresses this question.** In Appendix C, our goal is to test whether tampering attacks improve input-space vulnerability estimation even when other input-space attacks are already available. To prevent the non-independence  of results from different training checkpoints from the same run  affecting our analysis, We perform k-fold cross-validation across model families by holding out all models trained by the same unlearning method, one method at a time. Figure 13 presents the final result, showing that tampering attacks help predict worst-case input-space vulnerabilities.
> >
> > ## What do you think?
> >
> > **Thank you again for taking the time to review the paper and providing helpful feedback! Do the above actions address your concerns with the paper? If not, what further clarification or modifications could we make to improve it?**

---

> > > ### Comment · Reviewer_rXbX · 2025-06-11
> > > **Reply**
> > >
> > > Thank you for your detailed reply, which addresses most of my concerns. I have several left writing-level suggestions based on the response.
> > >
> > > - **Question 1.1**: I recommend placing the main results in Section 4.2 of the main paper, as it is essential to demonstrate the generality beyond the results from the WMDP-bio dataset. Currently, all the results are put in the Appendix, which is not easy for the reader to refer to and make the main body self-contained.
> > > - **Question 2.1**: I agree that unlearning is a representative defense, and the evaluation in the scope of unlearning is extensive. However, other safety alignment techniques exist, e.g., RLHF in [r1]. Thus, that's why I suggest clarifying the scope more clearly as the author did in Action 2.2.
> > > - **Question 3.1**: I suggest that the author include potential ideas in the paper for future exploration.
> > >
> > >  - Another suggestion is to clarify the motivation of choosing unlearning or safety fine-tuned in the introduction (quoted below), as it's unclear to the reader why you are choosing LLMs after defense rather than vanilla LLMs, as it has never been mentioned before:
> > > > *"To answer these questions, we pit state-of-the-art methods for unlearning and safety fine-tuning in LLMs"*
> > >
> > > ---
> > > **Reference**
> > >
> > > [r1] Chittepu, Yaswanth, et al. "Reinforcement Learning from Human Feedback with High-Confidence Safety Guarantees." Reinforcement Learning Conference.

---

> > > > ### Author Response · Authors · 2025-06-12
> > > > **Thanks!**
> > > >
> > > > Thanks for the reply.
> > > >
> > > > * **Question 1.1:** Thanks. Because TMLR gives us the space, we are moving Figure 15 to Section 4.2 to present the successes of the 11 attacks against the suite of 9 refusal-trained models.
> > > > * **Question 2.1:** Thanks! We are adding a discussion of methods that train networks to act desirably like RLHF and contrast them with the unlearning methods that we focus on which train networks to not act undesirably.
> > > > * **Question 3.1:** We think this is a good idea. We added to our discussion section a discussion of some of our current work on improving tamper resistance including methods for (1) filtering the models pre-training data (2) training the model on incorrect information about the unwanted knowledge domain, and (3) using combinations of state-of-the-art unlearning techniques.
> > > > * **Additional suggestion** This is easy for us to add. On it.
> > > >
> > > > Thank you for the time and thoughts!

---

### Review · Reviewer_J9HP · 2025-05-14

**Summary Of Contributions:**

This paper proposes model tampering attacks (modifying weights/activations) as a complementary method to input-space attacks for evaluating LLM risks.  The experiments benchmark various unlearning methods against both input-space and model tampering attacks, focusing on unlearning dual-use biology knowledge. Key findings include: the identification of a low-dimensional robustness subspace, the ability of model tampering attacks to predict the success of held-out input-space attacks, and the observation that state-of-the-art unlearning methods can be easily reversed.

**Audience:**

Yes

**Broader Impact Concerns:**

This work provide a safety evaluation method, which may be employed with malicious users to attack LLMs.

**Claims And Evidence:**

Yes

**Requested Changes:**

- Consider more models and more metrics.
- Show the benchmark result of open-sourced LLMs in your main paper.

**Strengths And Weaknesses:**

### Strength

- Safety evaluation of LLMs is an important question.
- Propose a novel benchmarking strategy using previously proposed tampering attacks.
- The baseline experiments are extensive and some findings are insightful.

### Weakness

- The experiments primarily focus on unlearning dual-use biology knowledge using a specific model (Llama-3-8B). At least another two different models and metrics should be considered.
- The main results only consider one model. Since it proposes a new benchmark method, it’s encouraged to report the performance of all mainstream open-sourced LLMs (e.g., Llama3, Qwen, DeepSeek) with your new method. It will benefit the community a lot.
- The findings regarding a low-dimensional robustness subspace have already been found by previous works[1].

[1] Assessing the Brittleness of Safety Alignment via Pruning and Low-Rank Modifications. ICML 2024.

---

> ### Author Response · Authors · 2025-05-25
> **Thanks + reply**
>
> We are thankful for your time and help, especially related to discussing Wei et al. (2024) and misuse potential. We were glad to hear that you found this work to be novel and insightful! Please see responses to key points below.
>
> ## 1. Models
>
> > The experiments primarily focus on unlearning dual-use biology knowledge using a specific model (Llama-3-8B). At least another two different models and metrics should be considered.
>
> **We agree:** We acknowledge that only working with Llama3-8B-derived models is a limitation, and we agree with the value of continued research on tampering attacks in other model families.
>
> **In defense of our approach – lots of experimental depth:** While we only worked with Llama3-8B-derived models, this choice helped us to have a high degree of experimental depth, working with 2 tasks (bio unlearning & jailbreak robustness), 8 capability suppression methods, 11 attacks, and a total of 74 models. Focusing on depth allowed us to have more rigorous insights and release our suite of 64 models. We would argue that this paper was reasonably scoped in that it took our team concerted effort over ~6 months to implement, tune, and run all of the attacks and defenses that we used. Duplicating our experiments with other model families would be a challenge because hyperparameters do not generally transfer well across model families. We would rather produce work that experiments thoroughly and rigorously with one model family than perfunctorily with multiple. **In your final evaluation, please weigh our investigation into a single model family against the high degree of algorithmic diversity in our experiments.** Overall, we would argue that the volume of experimentation and insights is more than enough to usefully contribute and meet a reasonable standard. Valuable papers often only experiment with a single model family. For example, in your review, you mention [Wei et al. (2024)](https://dl.acm.org/doi/10.5555/3692070.3694226) who only experiment with Llama2 models and were published at ICML. Unlike our paper, [Wei et al. (2024)](https://dl.acm.org/doi/10.5555/3692070.3694226) only experiment with one task  – safety (via AdvBench) – while we experiment with two – biology unlearning and jailbreak robustness. We also note that they only experiment with the base models – not any unlearning or refusal post-training methods.
>
> **In followup work, we are working with OLMo-2 and SmolLM models.** In our current work, we are not exactly replicating what we did for this paper, but it is focused on substantially improving the state-of-the-art for tamper robustness. Our current work on this focuses on the OLMo-2 and SmolLM model families.
>
> **Don’t forget about the jailbreaking experiments.** Our focus is not just on unlearning dual-use biology knowledge. Section 4.2 is entirely about experiments with jailbreak robustness. The task, models, data, and algorithms used in experiments in 4.2 are different than in 4.1.
>
> **Question 1.1: What do you think of the jailbreak experiments from 4.2?** Do you think we should work to present 4.2 more centrally in the paper?
>
> ## 2. Wei et al. (2024)
>
> > The findings regarding a low-dimensional robustness subspace have already been found by [Wei et al. (2024)](https://dl.acm.org/doi/10.5555/3692070.3694226)
>
> **Thanks for pointing this out. We agree, but our experiments go a further than theirs.** At a high level, [Wei et al. (2024)](https://dl.acm.org/doi/10.5555/3692070.3694226) do something similar in that they look at the relationships between 4 types of tampering interventions and 3 types of input space attacks. However, they only analyze pairwise relationships between attacks rather than analyzing the rank of the subspace that the result span. We also hierarchically cluster attacks to study families.
>
> **Action 2.1: We added discussion of [Wei et al. (2024)](https://dl.acm.org/doi/10.5555/3692070.3694226) to our discussion of related works and robustness subspace experiments.**
>
> ## 3. Malicious use
>
> > This work provide a safety evaluation method, which may be employed with malicious users to attack LLMs.
>
> **Action 3.1: We updated the impact statement** Thank you for pointing this out. Because the tampering attacks that we study have all been researched and used previously, we do not expect our paper to be a marginal asset to anyone seeking to misuse an LLM. Model tampering evals also offer a defender’s advantage for closed-weight models because downstream users cannot implement them. We updated our impact statement to discuss these points.
>
> ## What do you think?
>
> **Thank you again for taking the time to review the paper and providing helpful feedback! Do the above actions address your concerns with the paper? If not, what further clarification or modifications could we make to improve it?**

---

> > ### Comment · Reviewer_J9HP · 2025-05-29
> >
> > Thanks for the authors' detailed response. I appreciate that some of my concerns have been addressed, including the task number, experimental depths and similar findings.
> >
> > Here are my comments about remaining concern:
> >
> > - I strongly suggest including at least one mainstream LLM, such as Qwen2.5. Since this paper is intended as a benchmark/evaluation study—as indicated by the title, abstract, and introduction—relying on a single model for experiments is insufficient. This contrasts with Wei et al. (2024), which presents more in-depth findings.
> >
> > - I recommand main stream LLMs like Qwen2.5 or even Llama-2, instead of OLMo-2 and SmolLM.
> >
> > - For Question 1.1, I believe the jailbreak task should be presented more clearly in the main text, rather than putting all results in appendix.

---

> > > ### Author Response · Authors · 2025-05-30
> > > **Thanks + reply**
> > >
> > > Thank you for the response. We appreciate your time and effort as a volunteer reviewer.
> > >
> > > * **Context and weighing breadth vs depth.** Thanks! We agree with the importance of working with multiple model families. Due to the work situation of our team and our focus on followup work, We regret to say that we will not be able to add experiments with Qwen2.5 or similar. But we hope that you weigh this weakness against the strength of the algorithmic diversity and depth of experimentation in your final review. We also ask that you consider this paper's experiments with Llama3 models in context of our current work on tamper robustness in SmolLM and OLMo2 models. Meanwhile, [Hofstatter et al. (2025)](https://arxiv.org/abs/2502.02180) also use fine-tuning attacks to identify trojans in Gemma and Mistral models.
> > >     * **Wei et al. just use Llama2.** We stress that [Wei et al., (2024)](https://arxiv.org/abs/2402.05162) only experimented with Llama2 models. In section 3.1 they write: "Our experiments use Llama2-7B-chat and Llama2-13B-chat"
> > > * **We are using OLMO2 and SmolLM because our work is investigating pretraining interventions.** Thanks for the comment. To elaborate on our last comment, our current work does not work with the canonical SmolLM and OLMo2 models but instead models trained from scratch with the same dataset and architectures. This is because we are training some of these ourselves in order to investigate how filtering bio content from the pretraining data affects tampering attacks that aim to elicit bio capabilities. The business of pertaining the models is a big undertaking. This is the reason why we are using these models.
> > > * **We moved jailbreak attack results to the main paper.** Thank you for the feedback! Because TMLR can accommodate the extra space, we are moving figure 15 to the main paper.
> > >
> > > Thank you!!

---

> > > > ### Comment · Reviewer_J9HP · 2025-05-30
> > > >
> > > > Thanks for the authors' reply. However, I believe that relying on the results from only one single model significantly weakens the main contribution of this work, which aims to propose a **evaluation** method. A rigorous evaluation should not be limited to a single model; broader validation across models is required.
> > > >
> > > > Additionally, as I mentioned in my previous comment, the situation in Wei et al. (2024) is different—they focus on **presenting an in-depth insight** rather than proposing an evaluation framework. Moreover, they consider two models with different sizes (Llama2-7b/13b), which strengthens the validity of their findings. You could consider including results from LLaMA3-14B as well.
> > > >
> > > > In summary, for a paper that aims to propose an evaluation method, it is essential to validate the approach on more than one model. If additional results using different models are included, I would be inclined to give a positive recommendation. You can test only a representative subset of attacking methods to reduce the burden.

---

> > > > > ### Author Response · Authors · 2025-05-30
> > > > > **Thanks**
> > > > >
> > > > > Thanks for the quick reply again.
> > > > >
> > > > > **Practically unable to add Qwen experiments:** Because of the funding and affiliation-related context behind the original work, we are not practically able to revisit this to run things on another base model like Qwen. However, we are confident that our current work on tamper resistance, discussed earlier, is the best thing for us to be working on now. And we are continuing to contribute to the growing evidence base around model vulnerabilities to tampering attacks in multiple model families.
> > > > >
> > > > > **Closing argument:** Our final thought to leave is that we think rejecting this kind of paper based principally on lacking experiments in more than one base model could potentially be somewhat of an artificial standard. For your consideration, our closing argument would be this:
> > > > >
> > > > > * **We do not think it is accurate to say that Wei et al. present "an in depth insight" but that this paper does not.** We think that in all significant ways but one, our paper conducts a more comprehensive investigation. Wei et al experiment with two Llama2-family base models. But our experiments encompass both jailbreaks and unlearning, uses more attacks, and uses more defenses.
> > > > > * **Lacking diversity in model family does not mean lacking diversity in model mechanisms.** Our use of 8 unlearning methods to investigate biology unlearning allows us to gather evidence across models that have different mechanistic implementations of unlearning. **In this respect, we experiment with significantly greater model diversity compared to a work like Wei et al who only use base models off the shelf.**
> > > > >
> > > > > It is possible that we misunderstand, but we would argue that Wei et al. would likly not clear the bar reflected in your comments.
> > > > >
> > > > > **In conclusion:** We hope that these thoughts are useful for your final assessment. We regret that we cannot add Qwen experiments due to funding and logistical reasons. And we recognize that this is a weakness of the paper. But we hope you share our excitement about our continued work on tamper resistance, and we hope that you consider our use of 8 unlearning algorithms in your final review.
> > > > >
> > > > > Most of all, thank you for the time and thoughts!

---

### Decision · Action_Editor_AUmn · 2025-06-19

**Recommendation:** Accept as is

**Audience:**

Yes

**Audience Explanation:**

Evaluating the safety of LLMs is an active research area and this work will certainly be of interest to researchers in this area.

**Claims And Evidence:**

Yes

**Claims Explanation:**

The paper proposes that model tampering attacks (namely, fine-tuning and latent-space activation attacks) should be used to evaluate LLM safety in addition to input-space attacks. They demonstrate the potential insights that can be derived from such an approach in an extensive evaluation on various tasks, attacks and defenses. Models involved in the experiments will also be released.

All reviewers agreed that the evaluation was comprehensive and could provide interesting insights. While some reviewers had concerns about the generalizability of the work given its focus on Llama-3-8B-Instruct derived models, they mostly accepted the authors' explanation that the restriction in model families enabled a deeper analysis. This limitation is also indicated in the paper. Thus, overall the paper's claims are supported, given the stated limitations.